

# Quasi-particle functional renormalisation group calculations in the two-dimensional $t - t'$-Hubbard model

Daniel Rohe⋆

Forschungszentrum Jülich GmbH, Jülich Supercomputing Centre,
Simulation and Data Laboratory Quantum Materials, Jülich, Germany

⋆ d.rohe@fz-juelich.de

## Abstract

We extend and apply a recently introduced quasi-particle functional renormalisation group scheme to the two-dimensional Hubbard model with next-nearest-neighbour hopping and away from half filling. We confirm the generation of superconducting correlations in some regions of the phase diagram, but also find that the inclusion of self-energy feedback by means of a decreasing quasi-particle weight can suppress superconducting tendencies more than anti-ferromagnetic correlations by which they are generated. As a supplement, we provide sample results for the self-energy in second-order perturbation theory and address some conceptual matters.

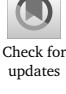

# 1   Introduction and motivation

**Background**

Quantum many-body systems of correlated electrons exhibit a wide range of different physical phenomena to be explored and understood. They have long been the subject of intense studies, and in recent times various advances in experimental as well as theoretical areas have been boosting this interest further. It is a ubiquitous and fundamental phenomenon in many-body physics and (quantum) field theory, that seemingly simple rules for microscopic constituents can lead to collective states and complex phenomena, which are entirely different from the underlying building bricks and can be considered in themselves as effective constituents, obeying different rules at a macroscopic level. This may even happen in several steps and over several scales. Bridging this gap and trying to identify mechanisms leading from microscopic "stand-alone" physics to macroscopic "collective" physics is a perpetual task. It requires not only constant progress regarding the models which can serve to describe real materials and effects therein, but also necessitates ongoing development, refinement and improvement of the methods that are available to treat such models mathematically.

In this context, the interest in the two-dimensional Hubbard model (2dHM) [1] as a prototype system and test ground for methods describing certain types of correlated electrons systems remains unbroken. Apparently simple in its definition, it can serve as a basis for the emergence of complex collaborative effects as a function of interaction, temperature, filling and band structure, with various types of correlations, ordering tendencies and kinds of non-Fermi-liquid behaviour competing and/or cooperating with each other. Even in the particle-hole-symmetric case at half filling there is to this day no method that can reliably reveal the physics of this model in the whole parameter range of interest. Rather, it has proven helpful to consolidate and connect results from various approaches in order to arrive at a more thorough and better understanding [2]. With the discovery of High-Tc cuprates, the attention paid to the model rose significantly, due to its potential relevance for understanding the underlying mechanism. We here revisit one such aspect of the model, namely the "coopetition" between anti-ferromagnetic and superconducting correlations. Functional renormalization group (fRG) methods have been an important tool for studying this aspect from early on [3]. Since then, fRG methods have been extensively applied to the 2dHM and to many other models, for a comprehensive review see e.g. [4]. While the fRG is often employed in a perturbative manner, and thus restricted to weak or at most moderate interactions, it has provided valuable information on correlation effects, often so in low-dimensional systems. It reveals potential mechanisms that can lead to superconductivity [3,5–7], captures the transition from anti-ferromagnetism

to superconductivity to ferromagnetism [8–11], and can be transferred to spin systems [12]. Generally speaking, it can extract the relative mutual strengths of a whole variety of correlations, that are associated with different types of order, amongst each other. It does so as a function of a continuous parameter and not as a single-step or iterative computation, and therefore sometimes allows to better follow and identify the underlying mechanisms.

Yet, many of the features of interest are non-universal in their behaviour, and the corresponding fRG results can depend on specific properties of the implementation - arguably an unfavourable property of any approach. This also concerns the inclusion and feedback of self-energy effects on the flow of the effective interaction. From a pure physical point of view, this is *a priori* likely to be a relevant ingredient for gaining a better understanding of the mechanisms. It accounts for certain changes of the underlying microscopic particles that are subject to mutual interaction, eventually leading to the collective effects of interest. During the fRG flow, this interaction feeds back on the constituent particles via scattering effects, which in a Fermi liquid-like picture primarily induce a finite lifetime and a reduced spectral weight of their coherent part, while spreading the remaining spectral weight over a wider energy range in a mostly incoherent manner. This feedback effect was neglected in many of the first approaches, mainly due to a considerable increase in computational demand. Intense efforts in method development and more efficient parameterisations of the relevant quantities allowed to address this matter in subsequent works within different approaches [10, 13–23], in which the influence of the gradual change of nature of the interacting particles was analysed from various perspectives. Some of these approaches are based on traditional RG regulators, such as a momentum or frequency/energy cut-off. This does not connect *different physical systems* during the flow, but approaches *the* physical system of interest in the limit of vanishing cut-off. The latter, however, is not always reached, by the very purpose of detecting divergencies in the flow and thereby instabilities. But that also means that self-energy effects near the Fermi surface - the relevance of which is of particular interest - lack the chance to feed back onto the flow. While this does by no means invalidate the approach, it motivates complementary schemes for which the fRG flow connects *physical* systems to each other. Such schemes are given e.g. by the temperature flow and the interaction flow.

Here, we use an extended version of the interaction flow method as a specific type of numerical treatment of fRG equations, first applied to the 2dHM in the particle-hole symmetric case at half filling in a previous work [24]. In a nutshell, it consists in a $Z$-factor-enhanced version of the original interaction flow method [11]. This provides conceptually simple means to include self-energy feedback in the coupled fRG flow equations for the self-energy and the effective two-particle interaction, while computing the self-energy directly on the real-frequency axis. The latter permits direct access to spectral features at a resolution of choice, which are not always evident to obtain or extract when working on Matsubara frequencies on the imaginary axis, in particular at weak coupling. Physically speaking, it accounts for a decrease in the quasi-particle weight, and the rate thereof, of the interacting particles and will be referred to as "quasi-particle enhanced fRG" (QP-fRG). As for the effective interaction, it neglects all frequency dependence, but does not rely on further approximations concerning the momentum dependence, such as channel decompositions [17] or truncated expansions/projections [25]. Rather, it is based on conventional patching schemes and provides a way to add self-energy feedback on top of the original "brute force" parametrisation as employed in early works. The additional *a priori* prospect of this extension was a potential improvement at the *quantitative* level. Results for the reference case at half filling and perfect nesting indeed suggested such an improvement [24], in the sense that the scales at which instabilities appear moved closer to results from other methods, a tentative improvement compared to fRG treatments based on bare propagators. However, here we move away from this reference case and find somewhat unexpected additional variations already at a qualitative level under certain circumstances.

Thus, rather than fully confirming previous results qualitatively and improving quantitatively, we found qualitatively different behaviour in certain cases.

**Outline**

This work is a direct follow-up of a work in which the method was first applied, namely to the half-filled and perfectly nested case [24]. We keep the overlap with this initial work at a minimum, mostly focusing on additional aspects. We will briefly recall model and method, to then treat a set of sample cases for which we present and discuss the results. We do also cover some matters which were not considered in [24], some of them in the appendix.

The content is to a substantial degree quite technical in nature. When it comes to extracting information about the physics, such aspects can be of relevant interest, and we observe that the type of fRG equations that are used for the self-energy feedback are decisive in certain ways. We will always compare computations for the case without any self-energy feedback to two different but common implementations of flows that do include the self-energy flow and its feedback on the flow of the effective interaction.

## 2   Model and method

The 2dHM is considered to be a prototype model in the context of correlated electrons, with a strongly revived interest since the early 1990s, largely triggered by the discovery of High-Tc cuprates. Lately, it has also been considered relevant for Nickelates in certain regimes [26]. Yet, the question in how far this highly reduced and idealised model can serve to describe fundamental physical aspects and mechanisms related to those in real materials remains in many respects open, or has not been answered beyond doubt. A general, exact solution is not available, and in particular away from half-filling and perfect nesting also solid numerical benchmarks are scarce, not to say not existent, owing e.g. to the sign problem in Quantum Monte-Carlo methods. Thus, improving and developing approximate methods remains necessary to gain further insight. This work constitutes an additional step in the treatment of the 2dHM by fRG methods in the regime of weak to intermediate bare onsite interaction. It shares the original motivation from prior works in that area to investigate the potential of the fRG to identify candidates for emergent properties of the model, with the intention to find ways of improving the quality of the approximation. A major difficulty that arises in this approach in the task to decide what is actually "better", since we cannot in general gauge the results against a known solution. We can however follow certain indications from other works and present results which can then be compared to other approximate methods.

**Model**

We consider the one-band Hubbard model on a square lattice for Spin-$\frac{1}{2}$ Fermions in two dimensions given by

$$H = \sum_{\mathbf{j},\mathbf{j}'} \sum_{\sigma} t_{\mathbf{j}\mathbf{j}'} c_{\mathbf{j}\sigma}^{\dagger} c_{\mathbf{j}'\sigma} + U \sum_{\mathbf{j}} n_{\mathbf{j}\uparrow} n_{\mathbf{j}\downarrow},$$

with a local interaction $U$ and hopping amplitudes $t_{\mathbf{j}\mathbf{j}'} = -t$ between nearest neighbours and $t_{\mathbf{j}\mathbf{j}'} = -t'$ between next-to-nearest neighbours on a square lattice. The sums run over all lattice sites $\mathbf{j}$ and spin indices $\sigma \in \uparrow, \downarrow$. The corresponding dispersion relation reads

$$\epsilon_{\mathbf{k}}^{0} = -2t(\cos k_x + \cos k_y) - 4t' \cos k_x \cos k_y,$$

and has saddle points at $\mathbf{k} = (0, \pi)$ and $(\pi, 0)$, leading to logarithmic van Hove singularities in the non-interacting density of states at energy $\epsilon_{\mathrm{vH}} = 4t'$. We measure kinetic energies with respect to the bare Fermi surface and thus use the definition

$$\xi_{\mathbf{k}}^0 = -2t(\cos k_x + \cos k_y) - 4t' \cos k_x \cos k_y - \mu\,.$$

Throughout the paper we fix the energy scale by setting $t \equiv 1$. While in our previous work we used $t' = 0$, we here move on to the case of $t' \neq 0$ and $\mu \neq 0$.

## Method

The general framework within which we operate are the functional renormalisation group equations for the one-particle irreducible (1-PI) correlation functions. For a general and comprehensive review see e.g. [4]. While this framework fixes the hierarchy of equations and the fundamental roles of the functions involved, it requires specific choices, steps and approximations when it comes to numerical implementations. Such technical elements are, in particular but without claiming completeness,

(i) The choice of the regulator in the bare quadratic part of the action.

(ii) The choice of the explicit truncation level and/or the loop order for the flow equations.

(iii) Type and granularity of the discretisation used for the flowing functions.

(iv) The type of self-energy feedback opted for.

(v) Treatment of the Fermi surface and the flow (or not) thereof.

(vi) The choice of quantities chosen for the analysis of physical aspects.

(vii) The choice of the criterion when the flow is stopped, due to entering the strong coupling region.

Within the QP-fRG implementation these choices and conditions are:

(i) The regulator is chosen as a homogeneous scaling parameter $g$ with initial value $g_0 = 0$. It enters the action via a scale-dependent bare propagator defined as $G_g^0 = g G^0$ [11, 27].

(ii) The truncation level is chosen - as in most cases - by setting the *explicit* calculation and consideration of the flowing six-point and higher correlation functions to zero. The loop order at the conceptual level is one-loop, although technically two-loop contributions arise by means of an additional differentiation.[1]

(iii) The frequency-dependent part of the self-energy (two-point function) is parametrised by calculating its imaginary part directly on the real-frequency axis at an intermediate resolution and subsequently employing a spline interpolation on a finer grid, to then compute the real part via Kramers-Kronig. The effective interaction (four-point function) is parametrised as a frequency-independent function of a finite number of discrete patches in momentum space, illustrated in Fig. 21 in appendix B.1.
Neglecting the frequency-dependence can become a troublesome approximation, as will be discussed below. It imposes itself in the QP-fRG mainly for reasons of feasibility and is a priori justified at weak coupling since there is no frequency dependence in the bare model. It can

---

[1]This may be viewed as a differential variation of a reinsertion procedure, as employed in [6, 28, 29].

however become relevant, up to a degree that it may invalidate the approach, as also discussed in the preceding work [24].

(iv) The level of self-energy feedback turns out to be a decisive choice. We include three levels: i) No such feedback, as in the original interaction flow scheme, ii) feedback according to the truncated fRG hierarchy, and iii) feedback according to the replacement first proposed by Katanin [30]. The latter induces an *implicit* inclusion of higher order terms of the hierarchy and provides a certain amount of one-particle self-consistency.

(v) The treatment of the Fermi surface is possibly the most delicate part of the method, at least conceptually. Technically, it is rather simple and happens implicitly. We here keep the Fermi surface fixed in the sense that all propagators appearing in the diagrammatic representation of the flow equation possess the Fermi surface of the bare system, and also its dispersion. The quasi-particle feedback extends the original calculation based on bare propagators "only" by the inclusion of a quasi-particle weight. We address some conceptual aspects of this in appendix C.
It would be desirable to contrast this strategy of fixating the Fermi surface to the case of fixing the chemical potential and computing the flow of the density. This is however not feasible in this setup, mainly since it would induce a continuous flow of the Fermi surface which would imply a dynamical adjustment of the discretisation and parametrisation of self-energy and effective interaction, similar in spirit to [31]. This seriously increases numerical effort and complexity, yet it might be possible to extend the method in this direction in future work.

(vi) The quantities we will mainly focus on are the critical scale at which the enhancement of the effective interaction becomes large, and the Eigenvalues of the effective interactions in certain subspaces which correspond to specific types of correlations. As for the self-energy, we focus on the flow of the quasi-particle weight.

(vii) The stopping criterion concerning the transition of the effective interaction to strong coupling is another subtle aspect. In some works it is chosen to be rather low, of the order of the bandwidth. This ensures to remain in the region of validity of the method, but can render it difficult to identify the physical aspects, in particular when it comes to phase diagrams and competing instabilities. On the other hand, choosing it higher allows to better access "ladder-like" divergent behaviour, but compromises on mathematical rigour. Here, we opted to stop the flow when the ladder-like increase exceeds a factor of 100. This is further specified below. The fact that this criterion is not well defined makes it difficult to compare results, even more so when other technical choices imply variations in the results, too.

We thereby continue and complement a previous study on the two-dimensional Hubbard model within a specific numerical implementation [24] and extend its application to cases away from half filling and perfect nesting. The main idea behind this particular scheme are: i) The fact that it is based on a perfectly flat regulator at finite temperature renders the fRG flow interpretable as a continuous increase of the bare interaction under certain conditions, hence the name *interaction flow*. ii) The frequency dependence of the self-energy can be calculated directly on the real frequency axis, which permits to iii) compute the flow of a quasi-particle weight along its proper definition and insert this in the flow of the effective interaction.

The diagrammatic representation of the QP-fRG equations is given in Figure 1 for the standard self-energy feedback and Figure 2 for the Katanin replacement. Details of the method are described in the initial presentation [24]. We here discuss several additional aspects and

subtleties, some of which become relevant only when moving away from the perfectly nested case. The interaction flow as such [11], as well as a self-energy feedback via quasi-particle weight(s) [13, 14, 16, 29], have been employed in various works before, but to the best of our knowledge not combined within the same implementation, and not based on a direct real-frequency treatment of the self-energy. Also, the QP-fRG permits to include the Katanin correction, which turns out to significantly influence the results for certain cases.

The standard fRG equations most often used in practice are obtained by truncating the equations after the level of the four-point function, i.e. by setting the six-point and all higher functions to zero at all stages of the flow. The implicit extension to replace the single-scale propagator in the flow of the effective interaction by the full scale-derivative of the propagator was first proposed by Katanin, a main motivation being a better compliance with Ward identities [30]. Later, this extension was investigated further and justified in more detail [32]. When applied to models with mean-field-type interactions it reproduces the exact solution, i.e. the respective self-consistency equations, and lifts the method from being non-self-consistent to a minimal degree of self-consistency. We here apply the Katanin extension in a wider context in the sense that it affects also the dynamical part of the self-energy, and it is for various reasons not *a priori* clear that it renders better results. We do this by conjecture with respect to other works that have shown to profit from it [33, 34] and have thereby indicated that it can be a favourable way to implement self-energy feedback. It has also become one of the standard extensions that is commonly used.

## 3 Remarks and limitations

Before presenting fRG results we address a few issues which arise as part of the approximation and which require some *a priori* discussion to be aware of certain limitations before interpreting data.

### 3.1 Coherent vs. incoherent contributions and total spectral weight

The strategy of including a quasi-particle weight on internal lines on the right-hand side of the flow-equation involves the following steps:

(i) Calculation of the imaginary part of the self-energy.

(ii) From this, calculation of the real part of the self-energy, followed by subtraction of the value at zero frequency on to keep the the Fermi surface fixed.

(iii) If (!) the resulting spectral function is reasonably well approximated by a Fermi-liquid-like Lorentzian plus incoherent background, calculate the quasi-particle weight.[2] Strictly, this is given as

$$Z_g = Z(g\,\Sigma_g) := \left(1 - \partial_\omega \mathrm{Re}(g\,\Sigma_g(\omega, \mathbf{k}))|_{\omega=\xi_{\mathbf{k}}^0}\right)^{-1} .$$

But as outlined in [24] and also discussed below, the derivative of the real part is approximated as a finite difference at some distance from the origin, since in the region of lowest energies the self energy does not fulfil Fermi liquid criteria in the strict sense.

---

[2]In the preceding paper [24] as well as previous drafts of this work we referred to a "Gaussian" shape. That is a misnomer. Expanding the self-energy on the Fermi surface in analogy to Fermi liquid theory yields a Lorentzian shape [35, 36].

(iv) Approximate the spectral function by its Lorentzian part, and further by a delta function times the quasi-particle weight, i.e. neglect the effects of damping/scattering rate. Thereby, one-loop terms are approximated by renormalised coherent contributions in the following sense (derivatives are omitted for simplicity):

$$G_g * G_g = (G_{FL} + G_{incoh}) * (G_{FL} + G_{incoh}) \approx G_{FL} * G_{FL} \approx G_{FL}^0 * G_{FL}^0 \,,$$

where $G_{FL}^0$ indicates that damping is neglected, i.e. the propagators behave like free propagators, but include the renormalisation of the weight.

This approximation is motivated by the fact that the divergences as familiar from treatments using bare propagators stem from ladder-like one-loop contributions based on perfectly coherent propagators. In physical terms, the QP-fRG approximation describes the one-loop flow generated by fully coherent one-particle excitations, which are in addition renormalised in terms of their scale-dependent quasi-particle weight (only). Incoherent combinations are neglected, the argument being that the corresponding part of the spectral function is smeared out over a wide frequency range and can thus be expected not to contribute substantially to divergent contributions and/or the change thereof.

In case a second type of *coherent* structure appears in the spectral function, or even replaces the Fermi-liquid shape completely, this procedure may become invalid, as also discussed in the next section. Here, we shall find only small residual coherent structures reminiscent of a slight dip in an otherwise essentially Lorentzian shape. These structures should not affect the flow significantly when omitted, since they are quantitatively small compared to the Fermi-liquid-like part.

We shall emphasise that the resulting full spectral function, as calculated from the flowing self-energy, remains indeed normalised to unity, at all scales. In other words, the quasi-particle weight acts as a filter on the rhs of the flow equation, extracting the coherent part of the one-loop contributions from the full spectral function for the purpose of calculating an approximation to the self-energy.

Further more, isolated satellite states above and below the band edges *in addition* to a quasi-particle peak can in principle appear in the familiar manner in SOPT and thus also in the QP-fRG, but only for very high values of the bare interaction, say beyond $U = 10$, and thus way outside the region of *a priori* validity. We do not observe this to happen up to the state of the divergence of the flow, or rather up to the stopping criterion. That, also, is an *internal* consistency check: If such states did appear at smaller $U$ due to the growth of the effective interaction - which was actually one of the tentatively anticipated scenarios - the method would become inapplicable beyond that point.

## 3.2 Self-energy in SOPT

As outlined in preceding works [37] and mentioned in the original presentation of the QP-fRG method [24], a Fermi-liquid-like description may be regarded as *a priori not valid* in the strict sense due to the "if" in point iii) above: For the 2dHM, already bare second-order perturbation theory (SOPT) leads to various non-Fermi-liquid effects in the single-particle spectral function, in particular for the perfectly nested case [37–41]. We here note in particular the appearance of a dip in the spectral function in a certain range of finite temperature and interaction strength [24]. Since the QP-fRG flow is based on a Fermi-liquid-like parametrisation of the spectral function and thereby of the scale-dependent single-particle propagators, it is mandatory to check and comment on the applicability of this strategy in the simpler but closely related SOPT. If it fails there, it also fails for QP-fRG purposes. We present sample data for

self-energy and spectral function, obtained from raw data of the imaginary part of the self-energy via subsequent Akima spline interpolation and numerical Kramers-Kronig calculation of the real part. The resolution of the raw data is $\Delta\omega_{low} = 0.004$ for the low-energy region $-0.2 < \omega < 0.2$ and $\Delta\omega_{high} = 0.2$ elsewhere. The value at $\omega = 0$ is included in the direct computation. The resolution after interpolation is $\Delta\omega_{spline} = 0.001$ everywhere. While the latter may seem a fine enough grid, we work at $T = 0.001$ and thus at the same scale. It is thus the upper limit for a suitable resolution in frequency when directly looking at spectral functions. For the QP-fRG purpose it is sufficient, since we require only a generalised slope of the real part of the self-energy at higher scales and for that purpose need to compute the real part during the flow only at two real frequencies [24]. We present sample results at various temperatures, for momenta on the Fermi surface near the anti-nodal point, in order to illustrate the concept and motivate the quasi-particle description. The actual QP-fRG results are quantitatively different, but qualitatively analogous.

The figures depict for each case in the top part

(a) real and imaginary parts of the self-energy at real frequencies

(b) a low-energy zoom thereof

(c) the imaginary part of the self-energy at Matsubara frequencies on the imaginary axis

and in the remaining part for three values of $U = 1$, $U = 4$ and $U = 8$

(d) the full spectral function, its Fermi liquid approximation, and the difference between the two

(e) a low-energy zoom of the latter

all for the point on the Fermi surface at the anti-nodal region, i.e. $\mathbf{k}_F = (\pi, 0)$.

Note that the self-energy is adjusted by a global shift to fixate the Fermi surface *before* the spectral function is calculated, and also in the plots for the self energy - c.f. appendix B. In case of half filling and perfect nesting this correction vanishes due to symmetry, but not in general.

### 3.2.1 Reference case at elevated temperature: $T = 0.2$, $t' = 0$, $\mu = 0$

We begin with the reference case of half filling and perfect nesting at an elevated but not exceedingly high temperature, the results of which are shown in Figure 3. This case illustrates in how far a Fermi-liquid-like parametrisation of the spectral function can be a sufficiently suitable approximation in a case where the self-energy clearly is non-Fermi-liquid-like by strict conditions. A (negative) peak-like structure at $\omega = 0$ is visible in the imaginary part of the self-energy on the real-frequency axis, but not evident on the imaginary axis at Matsubara frequencies. Thus, had we worked at imaginary Matsubara frequencies we would not have access to this feature. This requires at least a computation on the continuous imaginary axis including the low-energy region [42].
At $U = 1.0$, the Lorentzian approximation closely matches the full spectral function, with a small dip-like feature at the peak position visible in the plot of the mutual difference. The weight under the Lorentzian curve is slightly reduced with respect to the full spectral function, consistent with the computed $Z$-factor, which is not evident by eye but checked and verified numerically.

While the applicability of SOPT is limited to small values of the bare interaction, it is still instructive to insert larger values of $U$ in these calculations, since this allows us to investigate in how far not only the *shape* but also the *magnitude* of the self-energy translate into spectral properties. Since in SOPT the self-energy factorises into an interaction-independent two-loop contribution that yields the frequency-dependence, and a pre-factor of $U^2$ that determines the overall magnitude, this possibility is inherent. We thus show data for the same parameters but larger values of $U = 4$ and $U = 8$ in the middle and lower part of Figure 3. We note that, when $U$ is increased,[3]

- The central peak becomes wider, as expected in a Fermi-liquid approximation and the resulting Lorentzian approximation of the spectral function.

- The shift of spectral weight to incoherent regions starts to become noticeable at $U = 4$ and is evident at $U = 8$.

- The dip-like structure that decorates the central peak becomes more pronounced for larger $U$, and larger in relation to the height of the central peak.

- Precursors of satellite bands start to develop at the band edges at $U = 8$. If we increase $U$ further, they will of course move out of the band eventually. To then detect them in (non self-consistent) SOPT we need to account for true delta-peaks. Yet, the appearance of truly detached states becomes visible quantitatively also when checking the sum rule under the full spectral function *within the band limits*, which starts to decrease from unity when such states develop. Since this effect sets in only for large bare couplings, it is of no further relevance for the QP-fRG flow.

For our purposes regarding the applicability of QP-fRG, it is important to note that the quasi-particle approximation of the spectral function seems a reasonable description in SOPT even at $U = 4$, and it is not invalidated *a priori*. Even at $U = 8$ it still reflects the main features, while it can of course not serve as a description for the split peaks that develop.

When moving away from half filling and perfect nesting, the parameter space we could cover is of course vast, and we will not engage in presenting comprehensive data here. Several aspects are however noteworthy when it comes to arguing about the validity and possible issues of the QP-fRG method, conceptually as well as numerically. For that purpose, we choose a set of parameters that escapes the delicate special case of perfect nesting at the van Hove level, but remains near half-filling. As in the main part, we set $t' = -0.2$ and $\mu = 0.4$, which yields a curved non-interacting Fermi surface that intersects the Umklapp surface about half way between the nodal and anti-nodal points. We see in Figure 4 that the features discussed above for the reference case become less evident, with the negative peak in the self-energy and the dip-like structure in the spectral function at low energies not being visible by eye, but still present when plotting the difference between the full spectral function and the Lorentzian approximation. Thus, the Fermi-liquid approximation improves when moving away from the reference case.

For these two examples, we chose an elevated temperature of $T = 0.2$ to illustrate the thermal origin of the negative peak in the imaginary part of the self-energy and the relevant non-Fermi-liquid features associated with it. In the main part of this work, however, we choose a much lower temperature, for which we will check the same aspects. In particular, at perfect nesting the self-energy becomes linear in the low-frequency region for $T \to 0$, and it is well-known

---

[3]SOPT, being second order in $U$, does not depend on the sign of the interaction. Thus, we should not interpret any of this as uniquely related to *repulsion*.

that this also affects Fermi-liquid behaviour in the strict sense of the definition [38–40], often associated with the concept of a marginal Fermi liquid, being distinct from the "thermal" non-Fermi liquid discussed above. Here, we shall check whether this seriously affects or even spoils the QP-fRG approximation.

In Figure 5 we show data for $T = 0.001$, again first for the reference case of perfect nesting. The imaginary part of the self-energy shows a nearly linear behaviour far into the low-energy regime within the numerical resolution, thus exhibiting everything but a negative-quadratic shape required for a strict Fermi-liquid. As a consequence, by means of Kramers-Kronig-relations, the real part has a slope that strongly varies in the low energy limit and becomes very steep. Also, for $U = 1$ the spectral function as such can only just be resolved by the numerical resolution we use. While this could be adapted and refined, we also note that the chosen approximation by a Lorentzian still coincides with the full spectral function at this resolution, the relative difference between the two being of the order of $10^{-3}$. For $U = 4$ and $U = 8$ this changes only slightly. For $U = 4$ and $U = 8$ the central peak broadens again, and the Fermi-liquid approximation actually becomes easier to verify numerically. In all, the quasi-particle approximation seems a reasonably valid option.

Finally, we repeat the analysis at $T = 0.001$ for a case away from perfect nesting, i.e. for $t' = -0.2$ and $\mu = 0.4$ and show the results in Figure 6. The self-energy is again more Fermi-liquid-like and the Lorentzian approximation to the spectral function actually improves compared to the reference case. In fact, now the imaginary part is very flat, with a small curvature at $\omega = 0$, inducing a nearly opposite effect compared to the nested case. This leads to very sharp central peaks even for larger $U$. Looking at the differences of spectral functions, we still note remnants of the dip-like feature, but only at very low scales, limited by the numerical resolution. Working at $T = 0.001$ is already a numerical challenge. We could further refine accuracy and resolution to better access asymptotic features, if we were to go beyond the purpose of validating the QP-fRG approach.

We emphasise that these are purely *numerical* checks. In SOPT, Fermi-liquid behaviour in its strict sense is in general not sustained in the 2dHM, neither for $T \rightarrow 0$ nor at any finite temperature, and at low temperatures we are trying to capture differences in delta-function-like objects. It serves best as an approximation at low temperatures and away from perfect nesting, i.e. strong enough frustration, which also favours the convergence of QP-fRG results as a function of the discretisation. The - physically motivated - idea of the approach is to do better than pretending that the constituent one-particles excitations retain their unit spectral weight all along the flow, but to instead parametrise their coherent part by resorting to an approximately suitable candidate in the "space of fully coherent Fermi-liquid-like quasi-particles".

## 3.3 Frequency-dependence of the effective interaction

In the preceding section we checked a necessary condition the self-energy has to fulfil for the approach to be reasonable. Other conditions concern the effective interaction. During the flow, i.e. with increasing bare interaction, the scheme becomes less accurate by construction, being perturbative in nature. In addition, the effective interaction can develop a substantial frequency-dependence, which is neglected within the QP-fRG. This frequency-dependence may lead to pseudogap-like features which can in turn invalidate the approximation of the spectral function by the combination of a coherent Lorentzian part and an incoherent background. In some perturbative and fRG approaches at small enough temperatures the onset of such a dip in the spectral function is indeed observable, albeit only in the very vicinity of critical behaviour and not in all fRG schemes [21, 28, 42–44]. At elevated temperature this effect can be more prominent [42], but then interferes with thermal effects already present in SOPT, as discussed in the previous section. To add to the issue, some of the relevant fRG schemes which

capture pseudogap-like aspects do not consider self-energy feedback [28, 42–44], while others do include it in terms of standard self-energy feedback [21]. That said, we can check the QP-fRG flow and the use of a frequency-independent effective interaction for internal consistency. But it seems quite clear that the onset of a divergence in any fRG flow signals the breakdown of a quasi-particle description, at least in exclusive terms. A maybe even more difficult situation arises when no divergence develops, a matter we will get back to in the conclusion. It would remain very desirable to add a frequency dependence to the QP-fRG scheme in the future, but that is a technically involved matter, and even conceptually not straight forward.[4]

We recall that, like other fRG schemes from which it is derived or to which it is intimately related, the QP-fRG can capture the onset of strong correlations and some of their implications upon approaching some cross-over scale $T^*(U)$ (or $U^*(T)$) from the normal, metallic phase, i.e. it is meaningful for $T > T^*(U)$ ($U < U^*(T)$). The divergences which appear in the numerical treatment near $T^*$ ($U^*$) are mean-field-like in nature, and the fRG in this scheme cannot flow into an "anomalous" phase where correlations and their effects are strong, but long-range order and symmetry breaking remain suppressed due to order parameter fluctuations. Concerning the question of a pseudogap in the one-particle spectral function *below* $T^*$, other methods such as e.g. the two-particle self-consistent approach (TPSC) [37, 46–48] and a self-consistent Ward identity approach [42] can give insight. They allow to investigate the effect of a frequency-dependent effective interaction in separate channels, which, loosely speaking, avoids long-range order and symmetry breaking while maintaining the near-critical properties of the interaction that trigger pseudogap physics in the one-particle spectrum.

## 4 Results for QP-fRG

In a previous, first application of the QP-fRG flow to the 2dHM we investigated the case of half filling and perfect nesting [24]. In that case, anti-ferromagnetic correlations dominate, with superconducting correlations building up in a subdominant manner. The scale at which a Slater/Thouless-like divergence in the flow of the effective (1-PI) interaction takes place is reduced upon inclusion of self-energy feedback. Since this feedback is achieved in the simplest manner by means of a quasi-particle weight, effects of quasi-particle damping, band structure renormalisation, and contributions involving the one-particle incoherent background are neglected. The main observation was a shift of the pseudo-critical scales, in general towards larger values of the bare on-site interaction, or equivalently to smaller temperatures.

Moving away from this special case of the half-filled Hubbard model alters the situation. We mainly varied the chemical potential $\mu$ and the strength of the next-nearest neighbour hopping amplitude $t'$, to some extent also temperature. This permits to tune the deviation from the nesting property and/or move the non-interacting Fermi surface away from van Hove singularities. In established fRG schemes, such parameter variations (can) lead to changes in dominant correlations, such as cross-overs from dominating anti-ferromagnetism to dominating d-wave superconducting or ferromagnetic correlations [4]. This picture has been consistently corroborated within fRG schemes that neglect all self-energy effects on the flow. However, in fRG schemes which calculate self-energy effects and *do* include the feedback thereof on the flow of the effective interaction, these findings persist and agree only partly and up to a certain extent, depending on specifics of the respective implementations and approximations [13, 15–18, 29, 49]. We will come back to this later, since we here observe yet another type of effect.

---

[4]During the revision of the manuscript a work appeared as a preprint on a more general formulation of a real-frequency dependence of also the effective interaction in fRG schemes [45].

Table 1: Specific cases for $\mu$ and $t'$ which are considered.

| $\mu$ | $t'$ | Description | Figures |
|---|---|---|---|
| 0 | 0 | reference case, half filling, perfect nesting | [7], [8] |
| $-0.1$ | 0 | hole-doped, away from half filling, no frustration | [9], [10] |
| $4t'$ | $-0.28$ | hole-doped, van Hove level below half filling, frustration via band structure | [11], [12] |
| $4t'$ | $-0.2$ | hole-doped, van Hove level below half filling, frustration via band structure | [13], [14] |
| $2t'$ | $-0.2$ | "mimics" half filling at weak coupling, frustration via band structure, intersections of Fermi surface with Umklapp surface | [15], [16] |
| 0 | $-0.2$ | electron-doped, frustration via band structure, Fermi surface touches Umklapp surface at $(\pi/2, \pi/2)$ | [17], [18] |

Most of the results presented in this work have been obtained for $T = 0.001$. This is in some sense the lowest temperature for which the numerics remain feasible and reasonably well controlled. At the same time, it is a low enough temperature to safely escape from purely thermal non-Fermi-liquid effects as they appear in the self-energy already in simple second order perturbation theory [37, 50]. Some results on temperature dependences are also included up to $T = 0.05$. In terms of granularity of the discretisation in momentum space, we mostly use a discretisation of 32 angular patches and seven slices in energy measured from the non-interacting Fermi surface, which was found to be a reasonable compromise in the half-filled nested case and corresponds to 224 patches in total. We did adapt the patching slightly with respect to [24] and use a finer patching resolution near $(\pi, 0)$, to account for the somewhat even more distinct role of the van Hove region away from perfect nesting. Some examples for the patching are shown in appendix B.1. We present results for a number of specific parameter sets, an overview of which is given in Table 1.

## 4.1 Reference case: $\mu = 0$, $t' = 0$

As a baseline, we present data for the half-filled perfectly nested case at $T = 0.001$ in Figures 7 and 8. The quantities we focus on are the flow of the effective interaction, an Eigenvalue analysis thereof and the flow of the quasi-particle weight.

### 4.1.1 Flow of the largest value of the effective interaction

We begin with the flow of the maximum value of the effective interaction $V_{max}(g) := max(|V_g|)$, where we omit momentum and spin indices. The flow is stopped if $max(|V_g|) > 100$, which defines the stopping criterion mentioned above. As shown in the left part of Figure 7, $V_{max}(g)$ develops clear and similar signs of divergence in all three cases under consideration: i) the case without feedback of the quasi-particle weight on the flow of $V_g$, ii) the case of standard feedback as it is derived from the original perturbative expansion/truncation of the 1-PI fRG equations, and iii) the inclusion of self-energy effects by means of the Katanin modification. The scales at which the flow diverges are larger for the two cases with self-energy feedback, while qualitatively the behaviour is very similar and quantitative differences are comparatively small. It is important to keep in mind that the definition of $V_g(U)$ is such that it represents the final result for the effective interaction at $g = 1$, while for all other values of $g$ it constitutes an isolated generalised Slater/Thouless enhancement. To obtain the "real" final result of the

effective interaction when stopping at an arbitrary scale $g$, one has to make use of the relation $V_{final}(g^2 U) = g^2 V_g(U)$ [11]. We do include this additional factor $g^2$ in the Eigenvalue analysis that follows below. In practice, we always set $U = 1$, use $g$ as the flow parameter and plot quantities as a function of $g^2$. This permits to set $g^2 = U$, by the very virtue of an interaction flow.

### 4.1.2 Flow of the largest Eigenvalues in anti-ferromagnetic and superconducting sectors

In order to extract information about the dominant correlations as related to specific ordering tendencies, in previous works mainly two approaches were used: a) by analysing specific elements of the discretised coupling function, or b) by calculating susceptibilities, either by means of their flow equation or a posteriori by means of their diagrammatic expression, see e.g. [5, 51]. Here, we choose an intermediate way, neither based on susceptibilities, nor on single components of the effective interaction. Instead, we compute the flow of the leading Eigenvalues and Eigenvectors in certain sectors of the effective interaction, in a similar fashion as they appear also in mean-field calculations that can be done in combination with fRG flows [52]. This analysis is limited in the sense that we do not explicitly compute susceptibilities and thus cannot compare their numerical values to other works. It is sufficient for our purposes here, since it allows to identify in which of the associated coupling channels a divergence appears and thereby which correlations dominate. The definitions we use for these Eigenvalues are given in appendix B.2. We mainly focus on the competition of (commensurate) anti-ferromagnetic and superconducting correlations. For the reference case at half-filling and perfect nesting we expect anti-ferromagnetic correlations to be dominant, and this is indeed what we see in the right plot of Figure 7 for all three different levels of self-energy feedback. The Eigenvalues include the factor of $g^2$ as described in the previous section and thus constitute the physical solutions at each scale $g^2 = U$. We again recall that the onset of divergences does not permit to identify a true phase transition. Order parameter fluctuations are not included in this description, and the dimensionalities of the order parameters which compete are not all identical.

### 4.1.3 Flow of the quasi-particle weight on the Fermi surface

Concerning the self-energy we focus on the flow of the quasi-particle weight as defined in [24] on the discrete patches of the non-interacting Fermi surface, see Figure 8. We notice an accelerated decrease due to the diverging effective interaction, while the values as such remain on a level close to one. Also, at this temperature and discretisation the Z-factor near the anti-node is smaller than near the node, with a monotonous increase towards the nodal direction. This is not fully the case for higher temperatures and finer discretisation, when thermal effects become more relevant [24].

## 4.2 Moving below half filling: $\mu = -0.1$ and $t' = 0$

We now set $\mu = -0.1$ and thereby shift the bare Fermi surface to a position below the Umklapp surface, defined via $k_x + k_y = \pi$, while keeping the next-nearest neighbour hopping at $t' = 0$. This removes the property of perfect nesting and moves the Fermi surface away from the van Hove level, thus reducing the density of states at the Fermi level. In direct comparison to the reference case we observe the following.

### 4.2.1 Flow of the largest value of the effective interaction

We see in the left part of Figure 9 that for the flow without any self-energy feedback the onset of the divergence is shifted to higher values of the bare interaction, to about $U \approx 1.8$. Including self-energy feedback in the standard form additionally shifts this onset to $U \approx 2.1$, maintaining the overall qualitative shape. For the case of Katanin feedback the difference is more significant and becomes qualitative in nature. The onset of a divergence disappears and the flow begins to flatten at about $U \approx 3$, still exceeding the chosen numerical limit at about $U \approx 3.4$.

### 4.2.2 Flow of the largest eigenvalues in anti-ferromagnetic and superconducting sectors

Similar to the reference case, the largest Eigenvalue consistently remains the anti-ferromagnetic one for all three cases, as seen on the right of Figure 9. However, the strength is reduced and the largest Eigenvalue in the (d-wave) superconducting sector builds up earlier and more substantially than it does in the reference case. This is in contrast to results in e.g. the Wick-ordered scheme at $T = 0$ with a sharp momentum cut-off [5], where superconducting correlations actually dominate in this region. This may however also be due to particular technical aspects of the Wick-ordered scheme used in [5]. Since fRG flows become less accurate when the flow develops a divergence, this contrasting behaviour is noteworthy but of limited significance. The main observation remains the build-up of superconducting correlations, triggered by the influence of structures in the particle-hole channel at intermediate scales of the flow.

### 4.2.3 Flow of the quasi-particle weight on the Fermi surface

The flow of the Z-factor differs from the reference case in certain aspects. While at the beginning of the flow there are little changes and the magnitude of the Z-factor(s) as such is very similar, the accelerated downturn which was apparent for the reference case is much less pronounced. This is already the case when no self-energy feedback is incorporated and the effective interaction does develop a divergence. The additional changes upon inclusion of self-energy feedback are rather small. Also, the spread between Z-factors along the Fermi surface is smaller, i.e. the values at the four patches are much closer to each other than in the reference case. We thus note that the self-energy affects the flow of the vertex more than vice-versa, at least on a qualitative level.

## 4.3 van Hove filling without nesting 1: $\mu = 4t'$, $t' = -0.28$

Next, we choose $t' = -0.28$ and set $\mu$ such that the bare Fermi surface touches the van-Hove points, i.e. we introduce frustration and remove the property of perfect nesting compared to the reference case, while maintaining a high density of states at the Fermi level. In previous fRG works it was found that in this range and without self-energy feedback anti-ferromagnetic and d-wave superconducting correlations compete at about equal strength for $U \approx 3$, with superconducting correlations likely prevailing in a low temperature range of $T \approx 0.001 - 0.01$ [7–9,11,15,17,51]. This is in accordance with the behaviour we find here in the case without self-energy feedback, but it changes when self-energy feedback is included.

### 4.3.1 Flow of the largest value of the effective interaction

On the left of Figure 11 we again show the flow of $V_{max}$. Similar to the case $\mu = -0.1, t' = 0$, the computation without self-energy feedback yields the familiar divergence in the effective interaction, at $U \approx 2.2$. When including the standard self-energy feedback, the divergence

persists but is shifted to higher values of $U \approx 5.6$, which has to be considered as outside the weak-coupling regime but may still be of some significance. We note in both cases a kink, which reflects the fact that superconducting correlations exceed antiferromagnetic correlations at this point of the flow. Again, the influence of the quasi-particle weight and its rate of change on the flow intensifies when the Katanin replacement is added. Then, the flow becomes regular without any signs of a divergence, and $V_{max}(g)$, i.e. the generalised Slater/Thouless enhancement, actually saturates. We show data up to $U = 9$, which is of course well outside the region of applicability of the method, but serves as additional technical information. (We recall that the effective interaction as such does not saturate but behaves as $\propto g^2 V_g = UV_g$. $V_{max}$ reflects an enhancement which is similar to the effect of the denominator in ladder approximations.)

### 4.3.2 Flow of the largest Eigenvalues in anti-ferromagnetic and superconducting sectors

On the right of Figure 11 the corresponding flows of leading Eigenvalues are again shown. Without self-energy feedback and for standard feedback, superconducting correlations exceed anti-ferromagnetic correlations at an advanced stage of the diverging flow, as anticipated from the kink in the flow of $V_{max}$. However, when including the Katanin replacement both Eigenvalues eventually behave linearly as function of $g^2 = U$, in accordance with the saturation of $V_{max}$. In particular, superconducting correlations remain much smaller than anti-ferromagnetic ones at *all stages* of the flow. This is a significant *qualitative* difference to the case without feedback and the case of standard feedback. It renders a different picture of the respective region in the phase diagram, where the dominance of superconducting correlations was typically found to prevail in prior works. It must however be stated that there remains the possibility of qualitatively relevant finite-size effects, as discussed below. Yet, what we consider to very likely be robust is the fact that the onset of strong correlations is shifted to bare interactions outside of the perturbative regime, and reliable statements on the outcome of the competition between the correlations are not possible.

### 4.3.3 Flow of the quasi-particle weight on the Fermi surface

The respective flows of the quasi-particle weights on the patches located on the non-interacting Fermi surface are very similar for the three cases, as shown in Figure 12. Similar to the reference case, the Z-factor decreases stronger near the anti-nodal direction than it does near the nodal direction. Already when standard self-energy feedback is included, the decrease for all Z-factors becomes slower and acquires an inflection point at about $U = 2.2$, in the sense that the second derivative changes sign. This is an additional feature, albeit appearing at larger values of $U$, not reached in the previous cases. While it is consistent with the saturation of $V_{max}$ for the Katanin replacement, it also appears for the case of standard feedback. There, the divergence is first hampered when $V_{max}$ behaves nearly linearly, and then restored in the superconducting channel.

## 4.4 van Hove filling without nesting 2: $\mu = 4t'$, $t' = -0.2$

The case $t' = -0.28$ was chosen since it can be compared to a number of previous works, and because it provides a setting in which the dominance of superconducting correlations was found to prevail in a robust manner. Yet, it is also not far from the tentative quantum critical point that marks the transition from AFM/SC to ferromagnetism, which in itself influences the flow and reduces the critical scales, already without self-energy feedback. In order to move further away from this transition point, we next show results for $t' = -0.2$ in Figures 13 and 14.

### 4.4.1 Flow of the largest value of the effective interaction

Upon reduction of frustration while staying at van Hove filling frustration, we expect the divergence in the effective interaction to develop earlier, that is at smaller values of $U$. This is indeed what happens for the case without feedback and the case with standard feedback, the transition being shifted to $U \approx 1.7$ and $U \approx 2.3$ respectively. For the Katanin feedback, $V_{max}$ still saturates, but at a much larger value compared to the case $t' = -0.28$. Also, it cannot be ruled out that finite size effects play a decisive role in this case, meaning that the saturating value may depend on the granularity of the momentum patching in such a way that the divergence may eventually get restored, as discussed below.

### 4.4.2 Flow of the largest eigenvalues in anti-ferromagnetic and superconducting sectors

Secondly, we also expect anti-ferromagnetic correlations to dominate over superconducting tendencies when frustration decreases. We do indeed note this shift in the Eigenvalues, however for the chosen stopping criterion the coupling channel associated with superconducting correlations still exceeds the one relevant for anti-ferromagnetism for the case without feedback and standard feedback. Had we however stopped the flow earlier we would assign this parameter set to the anti-ferromagnetically dominated region [11]. For the Katanin feedback, superconducting tendencies remain suppressed, but again it cannot be ruled out that finite size effects in the patching granularity shift the results for much finer discretisations.

### 4.4.3 Flow of the quasi-particle weight on the Fermi surface

As for the flow of the $Z$-factors, there are no decisive differences in comparison to the case $t' = -0.28$.

Overall, comparing the two cases $t' = -0.2$ and $t' = -0.28$, the question arises whether for the Katanin replacement the much stronger saturating behaviour for $t' = -0.28$ could be induced by the self-energy in the sense that its feedback shifts the system towards the ferromagnetic transition and thus to smaller critical temperatures. However, we did not notice any indications in that respect. In both cases, couplings in the ferromagnetic sector remain subdominant, and we do not include any renormalisation of the band structure, i.e. there is no effective change of the frustration which appears on internal propagators, which would of course add to the matter, as shown e.g. in [18]. As also mentioned above and discussed below, to us it rather seems to be an inverse logic: The Katanin replacement can lead to a saturation that converges in the patching when the bare system is chosen far enough from the reference case of perfect nesting and van Hove filling. This can be done by changing either of the two conditions, or both simultaneously.

## 4.5 Near half filling: $\mu = 2t'$, $t' = -0.2$

The choice $t' = -0.2$ and $\mu = 2t'$ creates a situation where a curved bare Fermi surface intersects the Umklapp surface roughly half way between the zone diagonal and the van Hove points. In terms of energy, the chemical potential is in the middle between van Hove filling and the case when the bare Fermi surface touches the Umklapp surface on the zone diagonal. While this does not *exactly* correspond to half filling for the free system, it does mimic this situation sufficiently well at weak coupling, meaning that we do not expect relevant variations of the results compared to an exact adjustment. This parameter set falls in the approximate nesting regime, as defined in [9].

### 4.5.1 Flow of the largest value of the effective interaction

Similar to the case $\mu = -0.1$ and $t' = 0$, the onset of the divergence in the effective interaction is shifted to higher values of $U$ for the case without self-energy feedback and for standard feedback, while the Katanin replacement again eliminates the divergence altogether, see the left plot in Figure 15. In contrast to the preceeding case of van Hove filling, there is *no* kink, suggesting that no change in dominant correlations takes place.

### 4.5.2 Flow of the largest eigenvalues in anti-ferromagnetic and superconducting sectors

The right plot in Figure 15 illustrates that in all three feedback cases correlations in the anti-ferromagnetic sector dominate over those in the superconducting one, in agreement with previous results [9]. In comparison to the reference case, superconducting correlations appear to come up stronger in relation to anti-ferromagnetic ones, while remaining subdominant. In some ways this is expected, since frustration and the resulting reduction of the nesting property disfavours anti-ferromagnetic tendencies. Yet, moving away from the van Hove level one may expect a disadvantage for superconducting correlations in the d-wave sector, which remains the leading channel.

### 4.5.3 Flow of the quasi-particle weight on the Fermi surface

The flow of the Z-factors along the Fermi surface is very similar for all three feedback cases, even at the onset of a divergence for the cases of no feedback and standard feedback, see Figure 16. The weight near the anti-nodal direction actually remains a little *higher* than between the anti-nodal and the nodal direction, as opposed to the reference case and the case of van Hove filling. In other words, the weight seems to decrease more near the point where the Fermi surface intersects the Umklapp surface, which is in accordance with previous findings and might be interpreted as reminiscent of a hot spot scenario [44]. A related variation at higher temperatures was also seen in [28]. The effect is however small here. It is consistent but should not be overly stressed. There are little to no signs of a steep decent for the two cases of diverging flows. For a thorough assessment of the hot spot scenario more detailed computations would however be required.

## 4.6 At the verge of Umklapp scattering: $\mu = 0$, $t' = -0.2$

The last set of parameters we present data for are $\mu = 0$ and $t' = -0.2$. In that case, the bare Fermi surface touches the Umklapp surface in the direction of the zone diagonal at $\mathbf{k} = (\pi/2, \pi/2)$. The results are a smooth evolution from the above case of quasi-half-filling.

### 4.6.1 Flow of the largest value of the effective interaction

In the left part of Figure 17 we note that the scales at which the flow transitions to strong coupling are further shifted to larger values of the bare interaction, and that the Katanin replacement once more suppresses divergences altogether. In the chosen range of $U$, the latter becomes less of a saturation effect and more of a sub-linear increase. From a purely numerical point of view, the saturation would set in at higher values of $U$.

### 4.6.2 Flow of the largest eigenvalues in anti-ferromagnetic and superconducting sectors

The right plot in Figure 17 shows again the flow of the Eigenvalues as above. It is noteworthy that there is little change compared to the case of quasi-half filling, also regarding the mutual

competition between anti-ferromagnetic and superconducting correlations. One could have expected that the superconducting tendencies are much less pronounced, given the fact that the lowest order d-wave form factor has little overlap with the remaining phase space for Umklapp scattering, the phases space for the latter being restricted to a narrow region around a few points only. What does change, though, is the shape of the Eigenvector, meaning that the overlap with the lowest order pure d-wave form factor becomes smaller compared to the cases above. A complementary analysis for the case without feedback and standard feedback suggests that at the transition to strong coupling the part of the particle-particle channel relevant for superconducting correlations decouples from other channels and begins to diverge, while this is not the case for the Katanin replacement.

### 4.6.3 Flow of the quasi-particle weight on the Fermi surface

The flow of the Z-factors becomes even more homogeneous than before and is essentially isotropic, as shown in Figure 18. Also, there is very little variation of the flow as a function of the feedback level. For the case with no feedback we note a faint sign of a steeper decrease for the patch in the nodal direction. This may here reflect the fact that Umklapp scattering remains active for this patch, while it is substantially more suppressed elsewhere.

## 4.7 Role of the patching granularity

Since we approximate the effective interaction by a discrete set of values on a finite number of patches in momentum space, we need to check in how far the results vary when we change the granularity thereof. Here, we find that it can have a crucial effect on the flow in the Katanin-extended version, while for the case without self-energy feedback and standard feedback it has much less impact. For the reference case, as it was also and originally treated in [24], a patching scheme of 32 angular sectors and seven slices in the non-interacting energy is sufficiently fine grained for the investigation of the critical value of $U$ at which the flow enters the regime of strong correlations. As it turns out here, a core reason for this resides in the fact that the flow to strong coupling sets in *early* enough, i.e. for comparatively small values of $g^2 = U$. Also, we cannot rule out that beyond the softly defined stopping criterion of $V_{max} = 100$ the Katanin feedback will lead to a saturation at very high values of $V_{max}$ also for the reference case, but that would be clearly outside the scope of applicability of the method as such, being perturbative in $U$ *and* $V$.

Conversely, when we move far enough away from the reference case, we also find it sufficient to use this level of patching granularity, but for different reasons: i) The flow of $V_{max}$ converges quickly enough as a function of the number of patches, but more importantly the flow in the Katanin scheme remains completely regular in $U$ even beyond the *a priori* regime of weak coupling, which we might in an optimistic manner extend to $U \sim 4$, but not really much beyond. We illustrate this in the left part of Figure 19 for the case of $\mu = 2t'$ and $t' = -0.2$, i.e. a sufficiently frustrated bare system at quasi-half-filling. We recall that the range in $U$ is extended to better describe and understand the numerical behaviour of the flow, but for physical reasoning we should to restrict ourselves to about $U \lesssim 4$. Even with this restriction, the inclusion of the Katanin replacement leads to significant *qualitative* changes. While without feedback and with standard feedback we have $V_{max} > 100$ for $U < 4$, it remains as small as $U < 10$ for $U < 4$ in the Katanin-extended flow. (We recall that $V$ is the enhancement factor, and the "true" effective interaction is given by $g^2 V$, such that eventually also this flow ends up at large values, but in a linear and trivial fashion, and not driven by a ladder-like divergence.) The intermediate region in parameter space between the reference case and cases far enough away requires finer granularities in order to decide which condition sets in first. In the right of Figure 19 a comparison is shown at a higher temperature of $T = 0.05$ between $t' = -0.1$

and $t' = -0.2$ for $\mu = 2t'$. Moving closer to the nesting situation and closer to the van Hove points leads to a situation in which the saturation depends substantially on the granularity, and a divergent-like behaviour is actually restored also for the Katanin replacement. In contrast, for $t' = -0.2$ convergence is reached already for $32 \times 7$ patches. This implies that we cannot choose a fixed granularity for parameter scans and from this compute a cross-over or phase diagram. Instead, we need to check for convergence at each parameter set. A full check of this convergence criterion is numerically expensive, but beyond the scope of this report. It is a conditional caveat when interpreting the results, and in itself a remarkable and subtle property of the coupled flow equations, which was unexpected. We noticed that a possibly similar phenomenon was observed in a field-theoretical treatment [14].

## 4.8 Cross-over at van Hove filling as a function of $t'$

A number of previous fRG works investigated the cross-over from anti-ferromagnetic to super-conducting to ferromagnetic tendencies with increasing $t'$ at van Hove filling [9,10,17,42,53]. Here, this picture depends on the approximation we use. While we observe the same cross over without self-energy feedback, and to a sufficient extent also when including the standard feedback, the Katanin extension as implemented here renders results that deviate from this scenario. As a consequence of the saturation effect, the cross-over sequence AFM-dSC-FM is no longer observed in the region $U < 3$ at the temperatures we could access. While technically we observe a saturation in the region of main interest, i.e. $-0.2 < t' < -0.4$, we avoid any physical statement beyond $U = 4$, since this is *a priori* out of scope for a weak-coupling method. Also, at van Hove filling the numerical delicacy is enhanced due to stronger angular dependencies and the high density of states at the Fermi level, even more so for smaller values of $t'$, as discussed above. We conducted fRG runs for a granularity of 528 patches, a practical limit set by the given compute conditions.[5] The left of Figure 20 shows results for the flow of $V_{max}$ at the selected cases $t' = -0.15$ and $-0.2$ and $-0.35$, for the flow without self-energy feedback and the flow including the Katanin replacement. In line with the specific cases above, the flow without self-energy feedback develops divergent behaviour in a region $U < 3$, while this is not the case for the Katanin feedback. For the latter, the flow again develops an inflection point and eventually saturates. We shall also mention that the inclusion of self-energy feedback in the standard way, i.e. without the Katanin replacement, leaves the qualitative behaviour unchanged compared to no feedback, but with the respective divergencies located at larger values of $U$. The latter can also exceed the weak-coupling region, but an inflection point never occurs and the flow always diverges at some point. In the right part of Figure 20, the ratio between the respective largest Eigenvalues in the superconducting and anti-ferromagnetic sectors are shown. It can be seen how superconducting correlations build up and start to grow over anti-ferromagnetic correlations for the flow without self-energy feedback, while in the Katanin version this ratio essentially saturates at a low value. We have also included data in the region where the fRG typically finds ferromagnetic tendencies. There, too, without self-energy feedback ferromagnetic correlations grow in a divergent manner, while in the Katanin scheme the flow saturates. There, superconducting correlations remain small compared to ferromagnetic ones, with the ferromagnetic instability not being present.

---

[5]Technically, we could further refine the granularity, however at high numerical cost that we could not account for at this stage.

# 5 Discussion and Outlook

In this work, we have applied a recently introduced extension of a particular fRG scheme, i.e. the quasi-particle enhanced interaction-flow method [24], to the two-dimensional Hubbard model away from half-filling and perfect nesting at low temperatures, presenting results for selected parameter sets. We find that the inclusion of self-energy feedback shifts the transition of the flow to strong coupling to larger values of the bare interaction, and for the Katanin replacement in some cases even suppresses all signs of a divergence. Upon inclusion of self-energy feedback, anti-ferromagnetic correlations are somewhat less diminished compared to superconducting correlations, in particular for the Katanin replacement, for the parameter sets that we considered. The flow of the quasi-particle weight is essentially smooth and regular, even just before the flow reaches the vicinity of a divergence. Clear signs of pseudogap behaviour are absent. At the technical level, we find that convergence in the granularity of the patching scheme is not homogeneous in parameter space, but varies substantially. It is well achieved sufficiently away from van Hove filling and for large enough frustration, but becomes numerically demanding and delicate in the intermediate region when moving towards perfect nesting. This is unfavourable, because that is a region of particular interest. Yet, this kind of numerical delicacy might reflect some of the physical complexity at work. Also, it should be feasible to better explore the convergence and finite-size aspects in the future, with compute power taking another major leap forward at this very point in time. As for the transition from anti-ferromagnetism to superconductivity to ferromagnetism at van Hove filling as a function of next-nearest neighbour hopping, we found qualitative agreement with prior results when no self-energy feedback was used and for standard self-energy feedback. Including the Katanin replacement, in contrast, leads to the disappearance of these transitions.

## 5.1 Comparison to previous works

Compared to various works on the role of self-energy feedback in fRG computations [10, 13–20], certain effects we found partially differ significantly upon inclusion of the Katanin replacement, while most aspects are consistent when the feedback is implemented in the standard way. Some of these prior approaches are based on self-energy feedback via a quasi-particle weight, as also done in the QP-fRG. A field-theoretic approach for the perfectly nested case showed saturation effects for the superconducting instability and a complete loss of the quasi-particle weight in the anti-nodal region [13]. Also in a field-theoretic approach, a saturation effect in the effective interaction was observed for the case of a flat Fermi surface away from van Hove singularities in [14], where a dependence on the granularity of the Fermi surface parametrisation was seen in a somewhat similar manner as for some cases in this work. The relevance of a two-loop approximation was discussed in [16], which also includes the self-energy feedback via a Z-factor, computed however on the imaginary axis and with a slightly different implementation of the single-scale propagator.[6] There, it was found that the inclusion of two-loop effects leads to a stronger decrease of anti-ferromagnetic tendencies than it does for superconducting ones, derived by means of susceptibilities. Other works included the self-energy feedback on the flow of the effective interaction by a direct calculation on the imaginary frequency axis. In [15], the changes to critical scales and the phase diagram as such, as compared to the simpler case without self-energy feedback, were found to be of secondary and only quantitative nature, while it was also concluded that the self-energy remains compatible with a quasi-particle picture. There, the Katanin replacement was *not* used, and these

---

[6]In the QP-fRG, two-loop diagrams also appear, but by other means. The QP-fRG conceptually works at the one-loop level and acquires access to two-loop contributions via an additional derivative with respect to the flow parameter.

results are in line with what is found here in QP-fRG for the standard fRG feedback. In [10], a related scheme was used, which *does* include the Katanin replacement, with a strong focus on the relevance of band-structure renormalisation, which we neglected here. There, it is also found that critical scales drop upon inclusion of self-energy feedback, but the dominance of superconducting correlations prevails, whereas we find it to disappear. This is maybe the most striking difference, albeit made by comparing schemes which are still very different in many respects. A similar route was taken in [17], where it was also found that the inclusion of a frequency-dependence in both, self-energy and vertex, still reproduces the dominance of d-wave superconductivity at van Hove filling. Another dynamical approach [18] suggests that superconductivity may be disfavoured in a broader range compared to the static case, c.f. Figure 8. therein. There, however, critical scales actually *increase* compared to the static computation, which is in contrast to our findings and assigned to the regulator.

The issue of regulator dependence, as mentioned in the introduction, hampers the comparison between fRG results, in some respects even at the qualitative level, but even more so on the quantitative level. This can be much improved within the multi-loop fRG approach as presented in [20], also including a comparison to the one-loop case with Katanin replacement. The inclusion of self-energy feedback shifts the increase of susceptibilities to larger values of the bare interaction, a direct comparison in terms of phase digram evolution is however not straight forward, since the focus of that work was different. In a brief and informal numerical comparison with external computations within a different technical implementation, the type of saturation effect we find here was not present, but possible signs of a similar effect were observed [54]. A systematic comparison of such kind might allow to further elucidate this matter.

In all, we find some general agreement but also some fundamental differences between previous studies on the role of self-energy feedback in fRG flows and the QP-fRG for the 2dHM at finite values of the next-nearest-neighbour hopping and different values of the chemical potential. At the same time, we also note that we observe the differences mostly *outside* of the (heuristic) region of perturbative validity of the fRG approach as such. Thus, we should also not over-interpret them. It also shall be recalled that numerical results of fRG flow equations sometimes depend significantly on details of the specific implementation, even more so for cases when correlations compete on comparable footing. Let alone the choice of a stopping criterion can alter things, in particular when comparing to results as we find them here, where an inflection point can occur. As a consequence, also the question whether anti-ferromagnetic or superconducting correlations are dominant is typically addressed by means of soft criteria. Several other aspects add to the variations, in particular the choice of the regulator and the parametrisation of self-energy and effective interaction and also the level of granularity when discretising the latter. In order to resolve some of these discrepancies it seems desirable to engage in a more direct numerical comparison under conditions which eliminate as much of these root causes as possible. In general, the multi-loop approach may constitute the most promising route within the fRG ecosystem to resolve the mutual discrepancies of previous computations [19, 20, 55–57], but it also requires a substantially higher effort than other approaches.

Under these conditions we consider the QP-fRG results presented here as indicative rather than conclusive, shedding light on the problem of interest from an additional perspective. The fact that a saturation of the flow sets in at all is an unexpected and noteworthy and new observation, which may be worth clarifying.

## 5.2 Caveats, limitations and additional remarks

We outlined certain aspects of the approximations which are made within the QP-fRG method in [24] and in the introductory part of this work. We here come back to some of these aspects

in light of the results, since they motivate a more specific discussion. We shall also offer some tentative thoughts which are to a certain extent speculative but might nourish some fruitful lines of thinking.[7]

### 5.2.1 Frequency-dependence of the effective interaction and pseudogap physics

Within the QP-fRG, the effective interaction is approximated to be frequency-independent. As discussed above, a number of works has added a frequency-dependence on the imaginary axis, with mixed conclusions. While in essence the current status quo can be summarised as a confirmation that the original calculations based on a frequency-independent effective interaction (aka. vertex) remain qualitatively unaltered, there remains the question in how far this is the case for the QP-fRG method, on a purely internal level and in comparison to other works. Considering the results that have been presented here, we note that for the cases where the effective interaction develops a divergence, the QP-fRG leads to a shift in critical scales and the relative strength of correlations, but not to a drastic deviation from the physical picture found in previous works. We note that in the initial stage of the flow, during which the mutual influence between particle-particle and particle-hole channels begins to induce the relevant structures in momentum space that first cooperate and later compete, the frequency-dependence of the effective interaction remains weak by perturbative arguments. At the advanced stage of the flow there are two distinct situation:

1. *If* the effective interaction develops a divergence, as familiar from essentially all previous fRG calculations, it is clear, or at least very much to be expected, that its frequency-dependence will eventually become relevant. However, there is no clear-cut criterion that permits to asses *at which stage* of the flow this becomes essential, in particular regarding its influence on the one-particle spectrum that in turn is crucial to be close enough to a Fermi-liquid-like description within the QP-fRG. To some extent this question has been addressed in methods that consider a frequency dependence of the effective interaction on the imaginary axis [21]. There, it was found that the self-energy *can* develop clear signatures of pseudogap behaviour in the single-particle spectral function when the flow enters far enough into the critical regime. Yet, while the appearance of a pseudogap - which is the main phenomenon which we have to worry about in QP-fRG when using a Fermi-liquid-like spectral function - is observed in the so-called Schwinger-Dyson version, it is however *not* clearly observable in the standard computation, as to be compared with the QP-fRG method, even when the effective interaction is allowed to flow to values of up to $10^3$. We thus note that even when a frequency-dependence *is* included in an fRG computation, this does not provide clear-cut evidence of the actual influence it has on one-particle quantities.

   Quite clearly, as stated in [24], *any* fRG flow will cease to render proper results when it enters a critical region of a divergent effective interaction, already for perturbative reasons and without worrying about the frequency dependence of the effective interaction. However, *when* and *how* that happens remains largely elusive. This can be regarded as one of the reasons why the stopping scales chosen for fRG computations do vary over several orders of magnitude, which constitutes another source of variation when trying to compare the mutual results, as mentioned above.

2. In contrast to the generic case of a divergent effective interaction, we have also observed cases where the flow does *not* diverge and the Slater/Thouless enhancement saturates.

---

[7]We recall that we restrict the discussion to two dimensions to avoid further complexity in the arguments, since in lower and higher dimensions other effects can arise.

For those cases, we consider the validity of the method to be restricted primarily on perturbative grounds, with respect to the bare as well as the effective interaction. A strong frequency dependence of the effective interaction *within* such limits, say values of $U \lesseqgtr 4$, is unlikely to develop, given that the dominant frequency-independent part remains finite and regular, not leaving much room for a strong or even singular frequency dependence. That said, the matter is in principle more subtle, since we cannot rule out that including a frequency-dependent effective interaction from the beginning would in turn restore a divergence. However, at least for the specific clear-cut cases of a saturation discussed and presented here, it seems unlikely that even if that was the case at all, it would restore the divergence at a scale below $U \approx 4$. There may of course be corner-cases and there certainly is a cross-over region between divergent behaviour and saturating behaviour.

These aspects need to be kept in mind when the results are compared to those of other sources.

### 5.2.2 Relation to the symmetry-broken case

The Katanin replacement is motivated by the fact that it improves the fulfilment of Ward identities in the truncated hierarchy of fRG equation for 1-PI schemes [30]. This led to the observation that it can connect fRG equations to mean-field solutions. More precisely, for certain reduced models, for which a mean-field treatment is exact, the Katanin substitution, in conjunction with an RPA-like treatment of the one-loop flow of the effective interaction, leads to an fRG flow that reproduces this exact solution when all modes are integrated out. In [32], this was shown formally as well as numerically, treating the specific example of a superconducting order parameter for a reduced BCS model. There, the Katanin replacement leads to the correct value of the gap in the fRG treatment. Its role is then discussed further by also providing results obtained within a standard computation using the non-modified fRG equations. For this case, a gap can still be found, but with a *lower* value compared to the exact result. This appears to likely be at odds with our findings, since in the QP-fRG the Katanin replacement leads to a stronger suppression of correlations compared to the standard case and thus to a lower mean-field-like critical temperature scale, which in analogy to BCS theory would translate into the opposite, namely a *smaller* gap in the ground state for the Katanin replacement, and not for the standard scheme. While we cannot identify a clear reason for this discrepancy, we consider the following aspects and differences to be potentially relevant:

1. In this work we do not treat a reduced model, but the "full" Hubbard model.

2. We do not restrict the one-loop flow of the effective interaction to RPA.

3. The regulators are different, which does not affect the final results in the exact computation, but may be relevant when the Katanin replacement is not employed.

4. The effects which enter the numerics and affect the transition scales are quite distinct: For the calculations in the reduced model, the superconducting gap is included, but no quasi-particle renormalisations. Here, it is the opposite.

5. In the computation of the mean-field-model, the Katanin replacement enters in the frequency-independent static part of the (off-diagonal) self-energy, while in the QP-fRG it enters via the dynamical part of the self-energy.

In all, we cannot fully and reliably clarify this issue and its potential relevance here. Somewhat related to this, we note that in [58], where a more general model, namely the attractive Hubbard model, is treated in the symmetry-broken state, the gap obtained via fRG and the Katanin

replacement is found to be *reduced* compared to the mean-field gap, see Fig. 16 in [58]. This reflects the fact that fluctuations which are not present in mean-field models can suppress the ordering tendencies. The value of the superconducting gap in the ground state was then investigated in more detail in a two-loop fRG scheme that extended the Katanin replacement and which includes a frequency-dependent self-energy and vertex, as parametrised on the imaginary axis [59]. There, the frequency-dependent scheme gives a lower value for the gap within the two-loop computation as compared to the one-loop case, while the purely static one-loop case yields an again smaller gap. However, no comparison to the standard fRG scheme is given in these works.

## 5.3 Outlook

While the QP-fRG as one particular implementation of a functional renormalisation group scheme provides some appealing features, such as direct access to one-particle low-energy features for real frequencies and the inclusion of self-energy feedback as calculated on the Fermi surface, it remains to be further compared and calibrated with respect to other methods in general and specific fRG implementations in particular. Due to the multiple possible choices and the non-universality of some of the quantities of interest in fRG approaches to the 2dHM, this seems a mandatory step for fRG implementations to advance on the quantitative level, as has been recognised and addressed in multi-loop approaches [19, 55–57]. An additional sensible test case might be the attractive Hubbard model away from half filling, since it eliminates some of the complexity of competing correlations and may allow to better disentangle certain underlying technical aspects when comparing different methods, as mentioned above for the symmetry-broken case. Some computations in this direction have been started within the QP-fRG framework.

## Acknowledgments

We gratefully acknowledge the computing time granted through JARA-HPC on the supercomputer JURECA [60] at Forschungszentrum Jülich. The author thanks Tilman Enss for valuable discussions, which led to the inclusion of the toy model for the Hartree shift. Also to mention is the usefulness of the C++ ODE-solver "ODEINT" [61], and support provided by Andreas Winter concerning the plotting software QtGrace. Credit also goes to the team that provides Jupyter-JSC as a dedicated interface to the supercomputers at JSC, which served well for part of the workflow and data analysis [62]. Computations were mostly conducted using the workflow environment JUBE [63].

# A Figures

## A.1 Diagrammatic representation of the flow equations

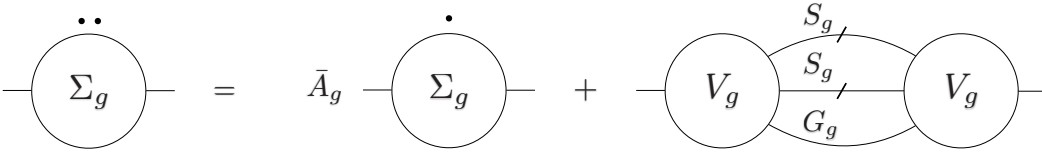

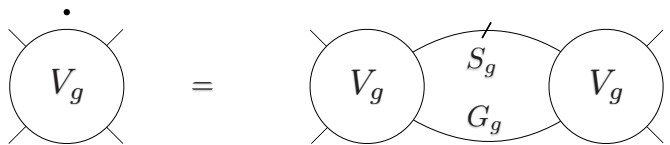

Figure 1: Diagrammatic representation of the QP-fRG equations, for the case of self-energy feedback based on the standard truncation of the 1-PI flow equations. The factor $\bar{A}_g$ is the average over the discrete Fermi surface patches of the quantity $A_g = 2\dot{Z}_g/Z_g = 2\frac{d}{dg}lnZ_g$.

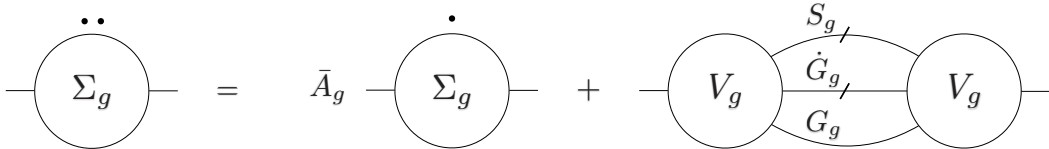

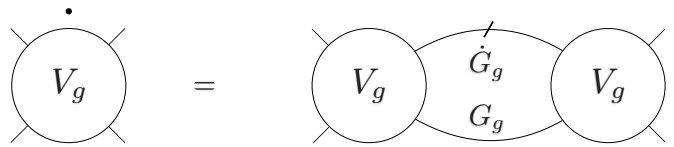

Figure 2: As Figure 1, for the case of self-energy feedback based on the Katanin replacement.

## A.2 Figures for SOPT

**SOPT for $T = 0.2, \mu = 0, t' = 0$ - near anti-nodal point**

Figure 3: SOPT results for the reference case $t' = 0$, $\mu = 0$ - i.e. at perfect nesting - and $T = 0.2$, on the Fermi surface and near the anti-nodal point, as described in the text. The real part of the self-energy is adjusted to fixate the Fermi surface and thus shifted by a constant to match $\text{Re}\Sigma(\omega = 0) = 0$.

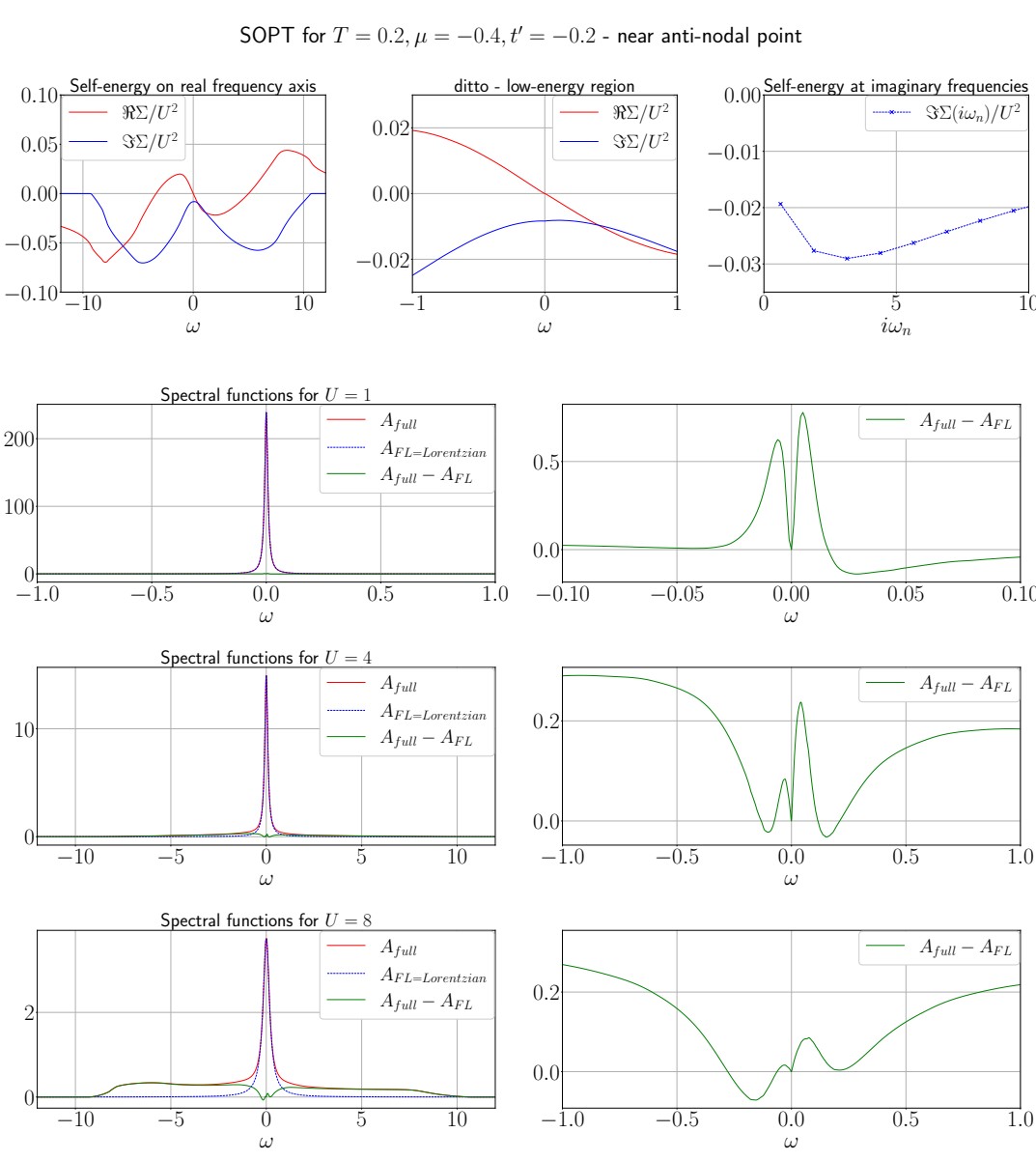

Figure 4: As figure 3, for $t' = -0.2$, $\mu = -0.4$ and $T = 0.2$.

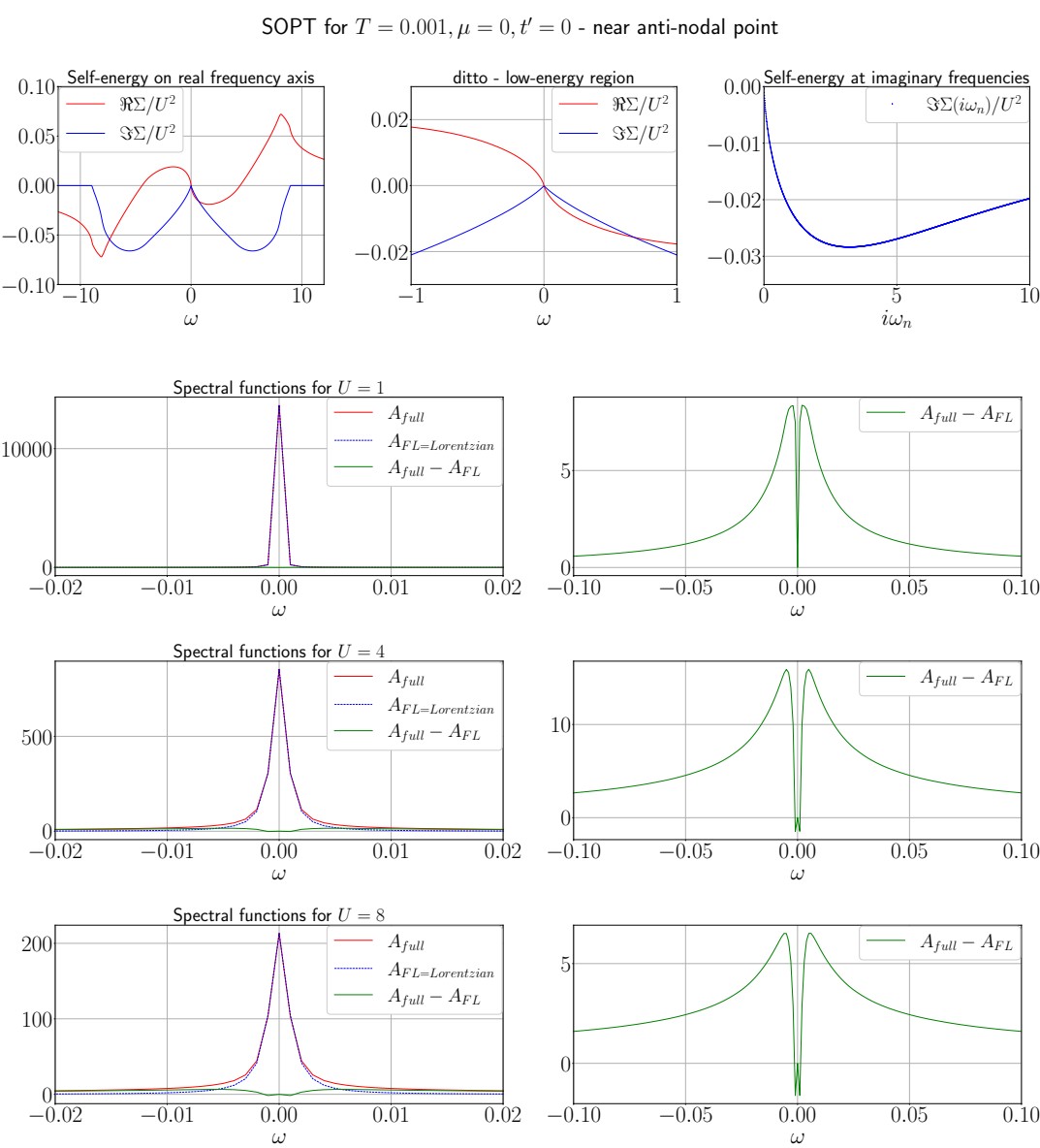

Figure 5: As figure 3, for $t' = 0$, $\mu = 0$ and $T = 0.001$.

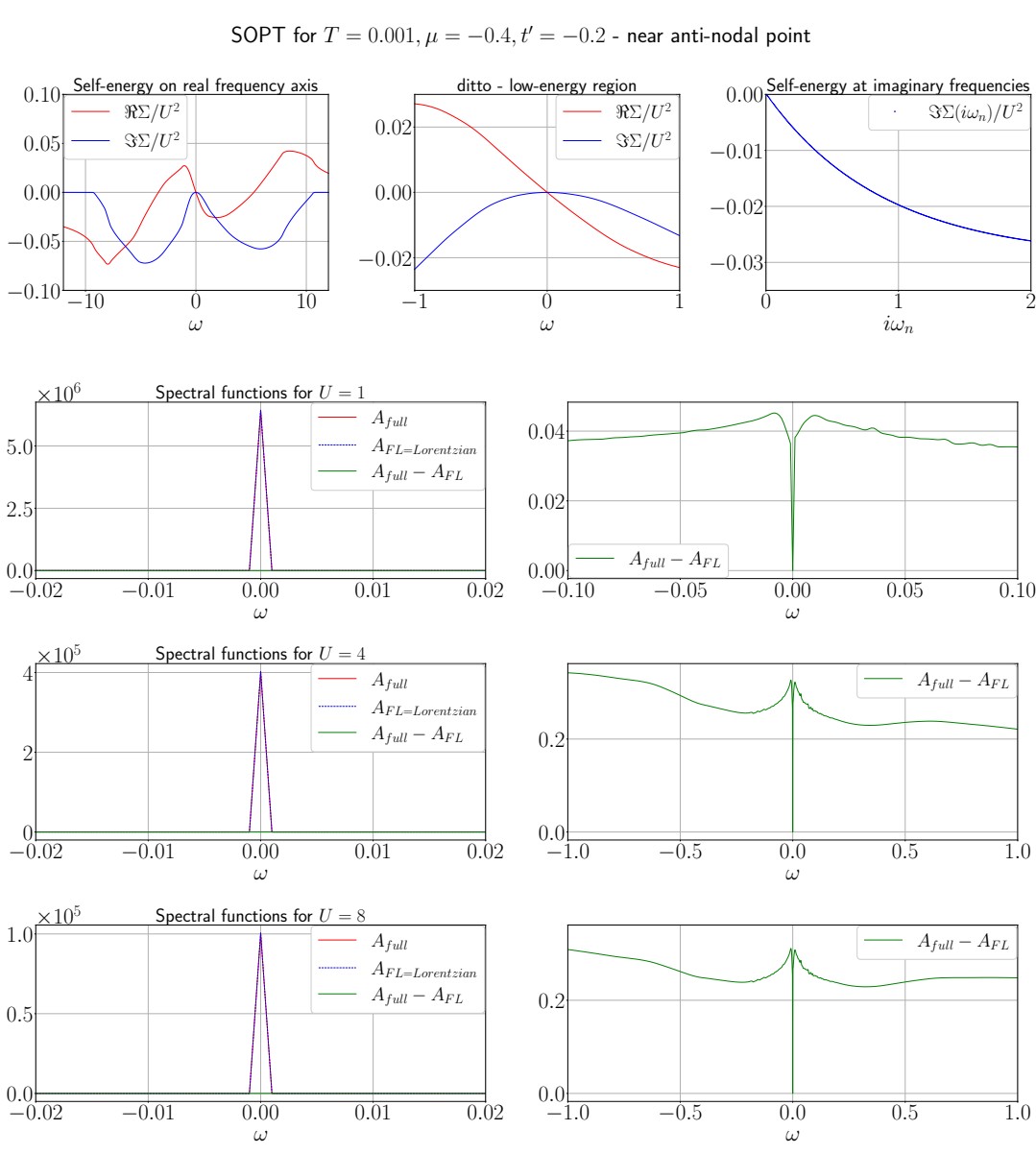

Figure 6: As figure 3, for $t' = -0.2$, $\mu = -0.4$ and $T = 0.001$.

## A.3 Figures for reference case - $T = 0.001$, $t' = 0$, $\mu = 0$

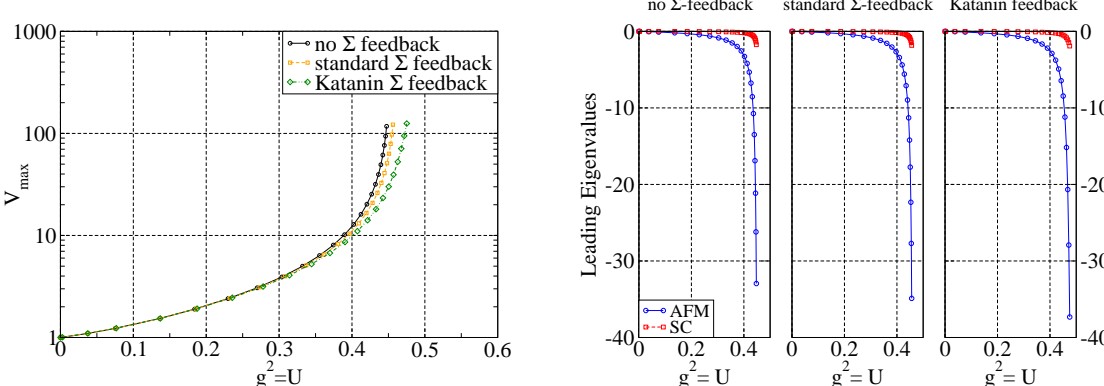

Figure 7: Reference case $T = 0.001$, $t' = 0.0$ and $\mu = -0.1$ , for the three versions of self-energy feedback. *left*: Flow of the largest value of the effective interaction. *right:* Flow of the largest Eigenvalues for the coupling sectors relevant for anti-ferromagnetic and superconducting correlations.

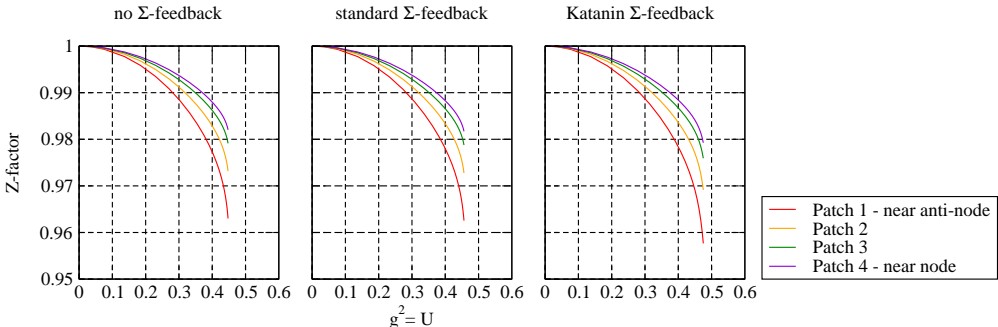

Figure 8: Flow of the Z-factors for the reference case for the three versions of self-energy feedback.

## A.4 Figures for $t' = 0$ and $\mu = -0.1$

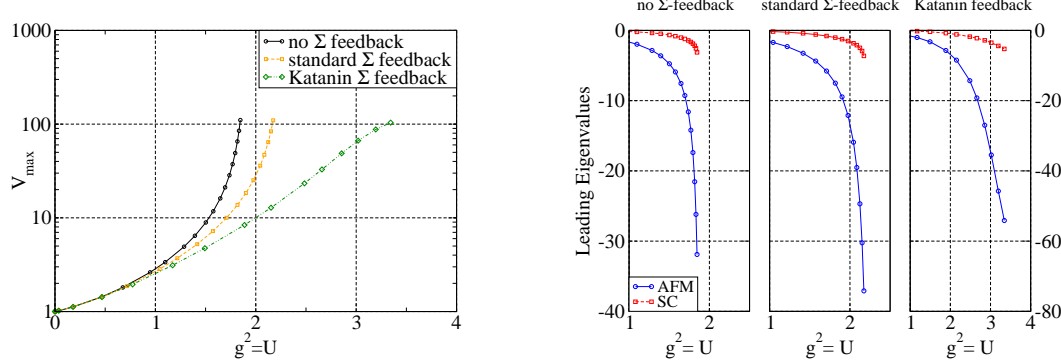

Figure 9: As Figure 7 , for $t' = 0.0$ and $\mu = -0.1$.

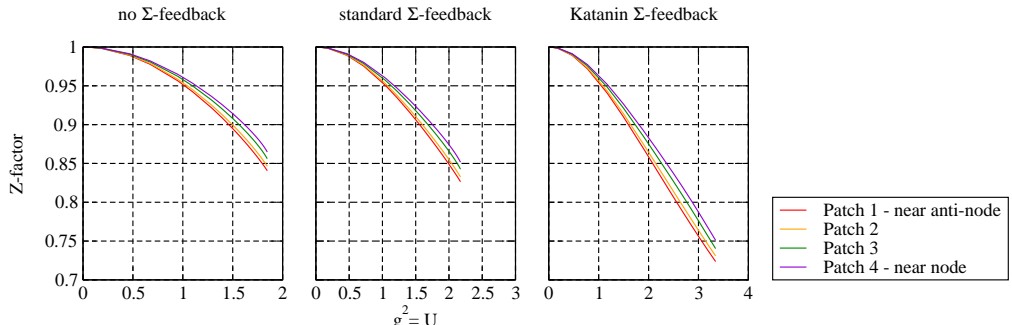

Figure 10: Flow of the Z-factors along the Fermi surface, for parameters as in Figure 9.

## A.5 Figures for $t' = -0.28$ and $\mu = 4t'$

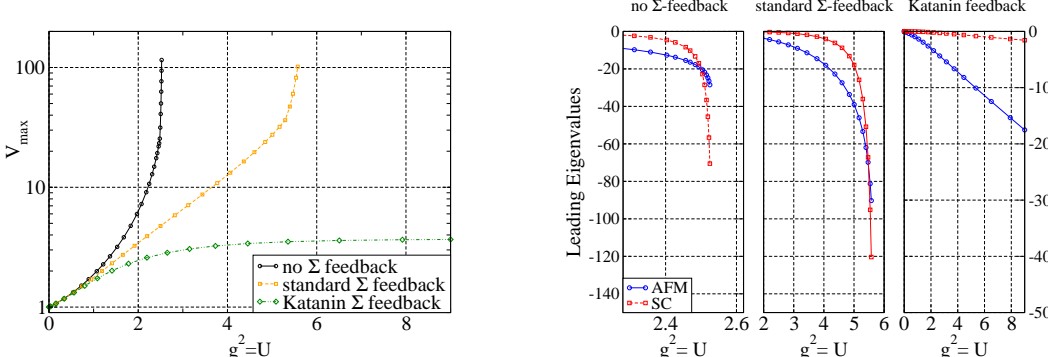

Figure 11: As Figure 7 , for $t' = -0.2$ and $\mu = 4t'$ - van Hove filling.

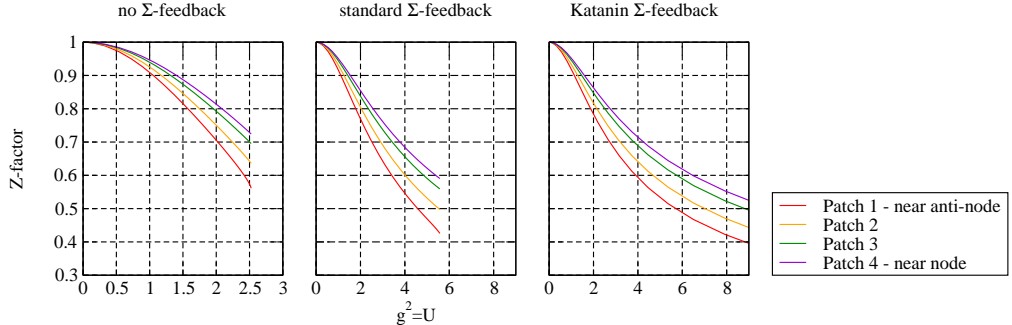

Figure 12: Flow of the Z-factors along the Fermi surface, for parameters as in Figure 11.

## A.6  Figures for $t' = -0.2$ and $\mu = 4t'$

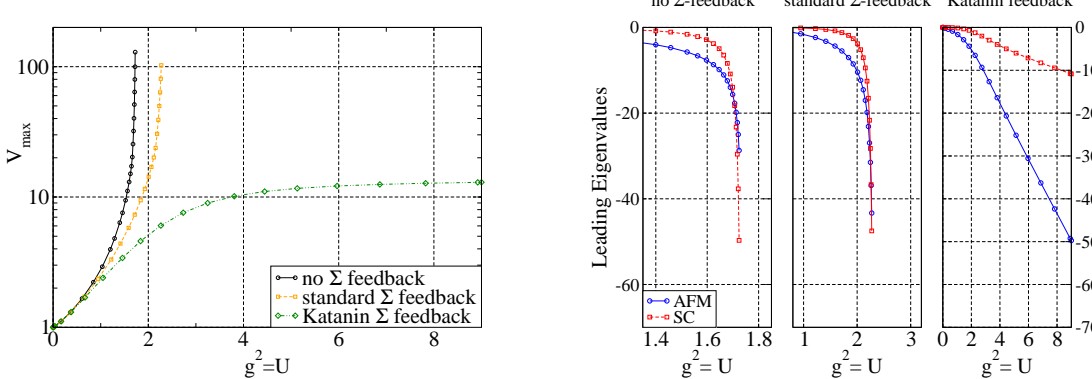

Figure 13: As Figure 7 , for $t' = -0.2$ and $\mu = 4t'$ - van Hove filling.

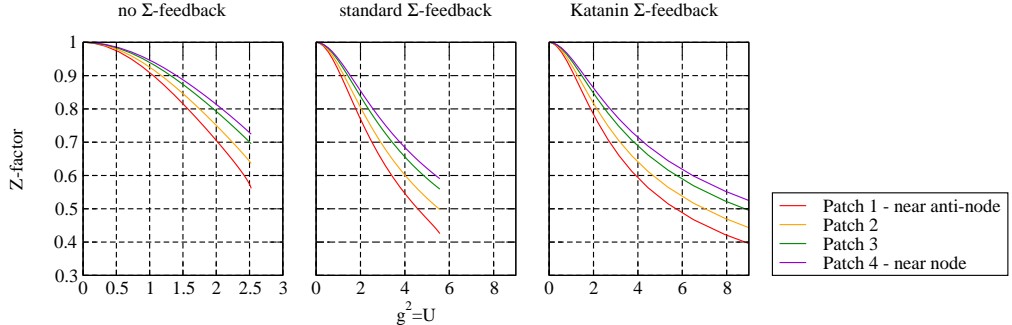

Figure 14: Flow of the Z-factors along the Fermi surface, for parameters as in Figure 13.

## A.7  Figures for $t' = -0.2$ and $\mu = 2t'$

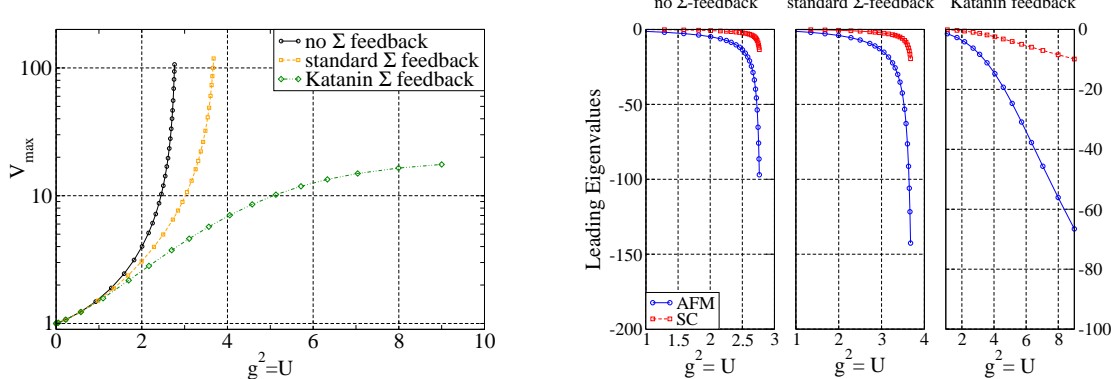

Figure 15: As Figure 7, for $t' = -0.2$ and $\mu = 2t'$ - near half-filling.

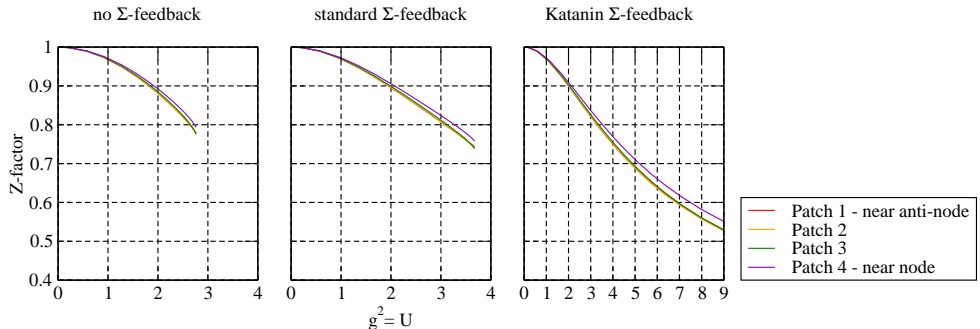

Figure 16: Flow of the Z-factors along the Fermi surface, for parameters as in Figure 15.

## A.8  Figures for $t' = -0.2$ and $\mu = 0$

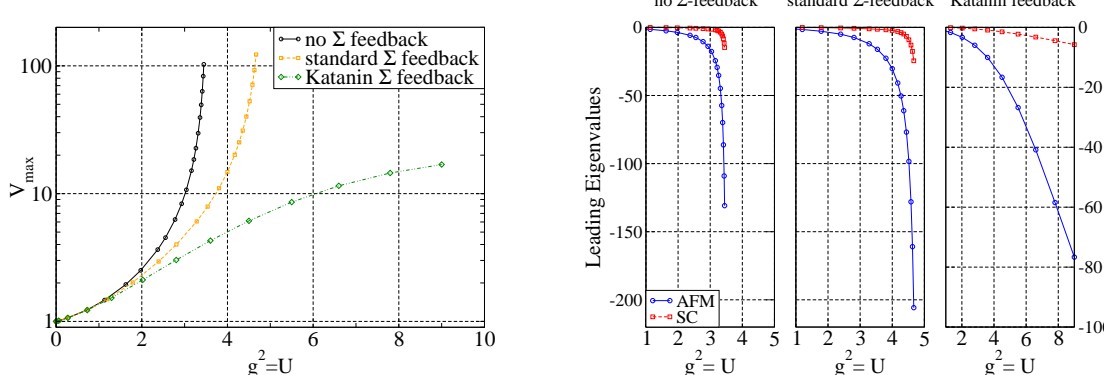

Figure 17: As Figure 7, for $T = 0.001$, $t' = -0.2$ and $\mu = 0$.

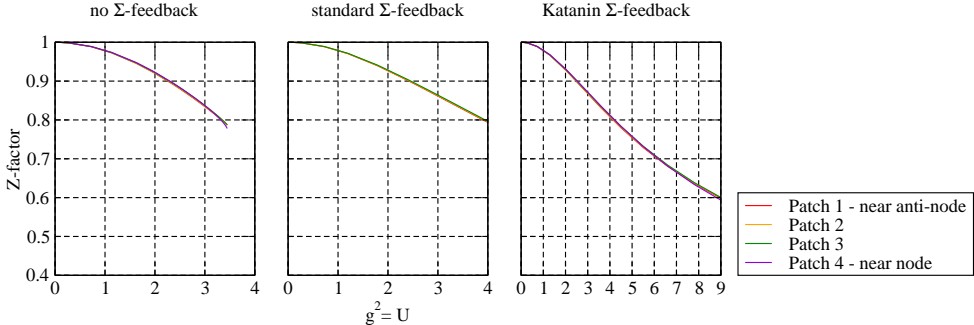

Figure 18: Flow of the Z-factors along the Fermi surface, for the same cases as in Figure 17.

## A.9 Figures for role of patching granularity

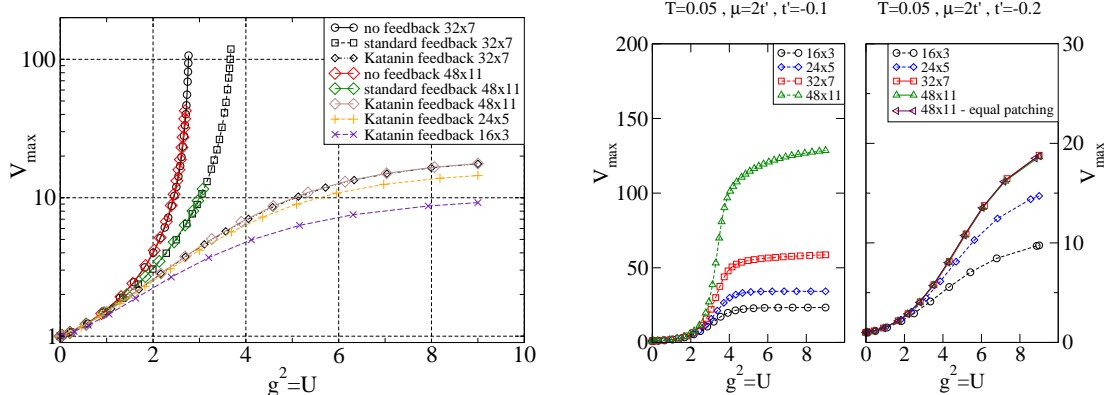

Figure 19: Dependence of the flow on the granularity of the patching. If not specified, the patching is 32 angular patches × 7 energy slices, as defined in the text. *Left*: Flow of $V_{max}$ as a function of patching granularity for $T = 0.001$, $\mu = 2t'$ and $t' = -0.2$. *Right*: Comparison of flows of $V_{max}$ for the Katanin extension as a function of patching granularity for $t' = -0.1$ and $t' = -0.2$ at $T = 0.05$, $\mu = 2t'$.

## A.10 Figures for the cross-over at van Hove filling

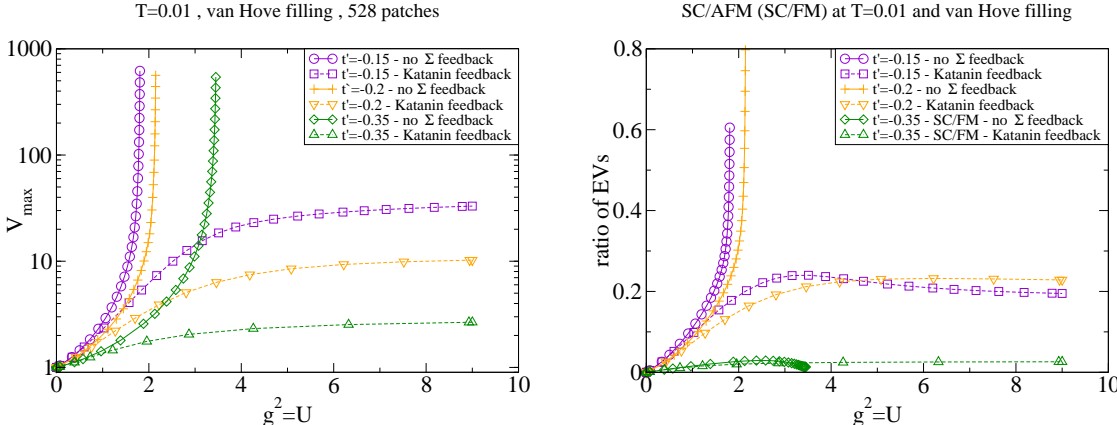

Figure 20: Comparison of self-energy feedback at van Hove filling as a function of $t'$ at $T = 0.01$. *Left*: Flow of $V_{max}$. *Right*: flow of the ration of Eigenvalues in the superconducting anti-ferromagnetic, respectively ferromagnetic Eigenvalues.

# B Supplementary material

## B.1 Patching scheme

In order to give an impression of the patching granularity, in Figure 21 we here show some cases of the discretisation in momentum space that was used. The grid is chosen finer towards the anti-nodal direction than in the nodal direction, to adapt to the variation of the local density of states, which becomes more relevant when frustration is included.

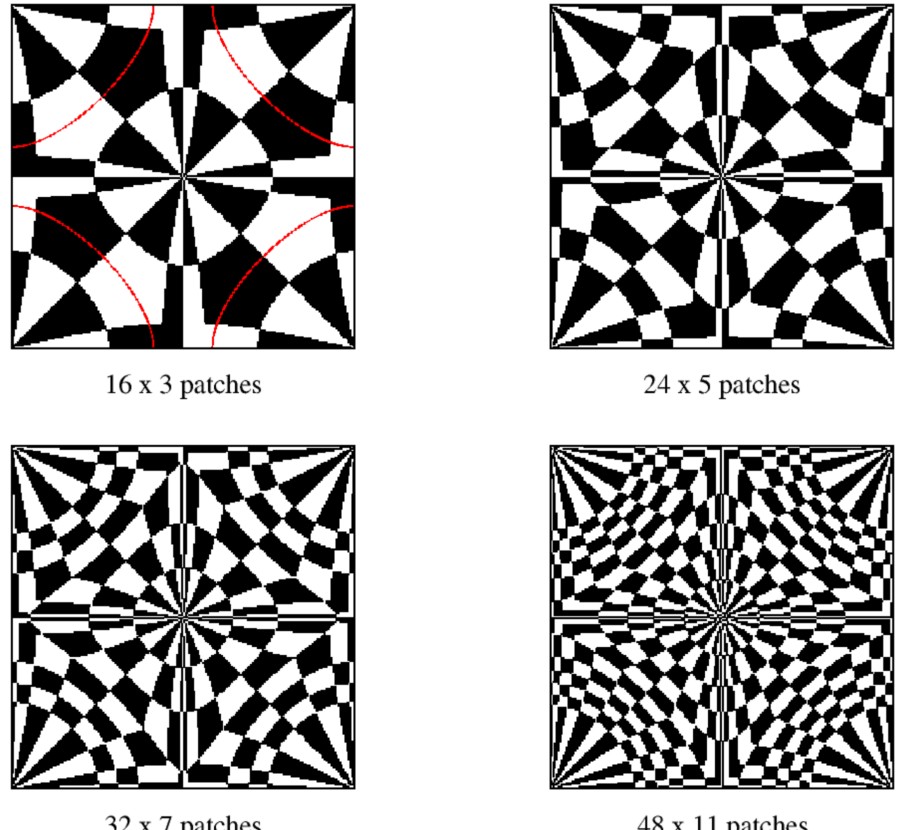

16 x 3 patches        24 x 5 patches

32 x 7 patches        48 x 11 patches

Figure 21: Patching granularity for different combinations of angular sectors and energy slices. We mostly applied the variant with $32x7 = 224$ patches.

## B.2 Definition of Eigenvalues/-vectors as relevant for certain correlations

The general conventions of parametrising the effective interaction are chosen to be in line with [5] and [52], splitting it in singlet and triplet parts $V^S$ and $V^T$. We then select the relevant combinations and subsets of the interaction corresponding to the respective ordering tendencies and compute the respective Eigenvalues and Eigenvectors. The Eigenvalues are normalised with respect to the size of the discrete matrices, i.e. the number of angular patches on the Fermi surface. The definitions (non normalised) we used are:

$$
\begin{aligned}
V^{SC}_{k_F,k'_F} &:= V^S(k_F,-k_F,-k'_F,k'_F) + V^T(k_F,-k_F,-k'_F,k'_F)\,, \\
V^{AFM}_{k_F,k'_F} &:= V^T(k_F,k'_F,k'_F+Q,k_F-Q) - V^S(k_F,k'_F,k'_F+Q,k_F-Q)\,, \\
V^{FM}_{k_F,k'_F} &:= V^T(k_F,k'_F,k'_F,k_F) - V^S(k_F,k'_F,k'_F,k_F)\,, \\
V^{CDW}_{k_F,k'_F} &:= 3V^T(k_F,k'_F,k'_F+Q,k_F-Q) + V^S(k_F,k'_F,k'_F+Q,k_F-Q)\,, \\
V^{POM}_{k_F,k'_F} &:= 3V^T(k_F,k'_F,k'_F,k_F) + V^S(k_F,k'_F,k'_F,k_F)\,.
\end{aligned}
$$

The definition of the scale-dependence within the interaction flow implies a flow of Eigenvalues as a function of $U$, as described in the main text and [24], given as

$$
V^X_{final}(g^2U) := g^2 V^X_g\,.
$$

We then use basic form factor functions which represent the symmetries of interest, onto which the Eigenvectors are projected, following [64]

$$
\begin{aligned}
\text{s-wave:} \quad & 1\,, \\
\text{p-wave:} \quad & \sin(k_x)\,, \\
\text{d-wave:} \quad & \cos(k_x)-\cos(k_y)\,, \\
\text{g-wave:} \quad & (A_{2g})\sin(k_x)\sin(2k_y)-\sin(2k_x)\sin(k_y)\,.
\end{aligned}
$$

These functions are normalised with respect to the discretisation granularity first and then used to project out the respective symmetries from Eigenvectors. While this does not account for all information contained in susceptibilities, it provides more information than single components of the coupling function.

## B.3 Scale invariance of QP-fRG equations

In the original set up of the interaction flow, the regulator is introduced as a homogeneous factor $g$ in the denominator of the bare quadratic action, i.e. in the numerator of the bare propagator of the non-interacting system, which starts from zero and continuously and homogeneously switches on all modes in the course of the flow. This can be interpreted as a continuous increase of the bare interaction in its exact formulation, which can be seen by means of the functional definitions and also in a simple manner directly in the truncated equations when self-energy effects are neglected [11]. When other approximations are applied, however, it is mandatory to check this property. This is what we do here, much analogous to the route in Appendix A of [21], to validate the following properties:

$$
\Sigma_{g/l}(l^2 U) = l \Sigma_g(U)\,, \tag{S0}
$$

$$
V_{g/l}(l^2 U) = l^2 V_g(U)\,. \tag{V0}
$$

For this to hold at all scales $g$ for an arbitrary auxiliary scale $l > 0$ we need to show that the derivatives with respect to $g$ are identical and that the initial conditions coincide.

As a preparatory step we look in more general terms at a function $f_g(U) = f(g, U)$:

$$
f_{g/l}(l^2 U) = l^n f_g(U)\,, \qquad \text{on the whole carrier for arbitrary } l > 0 \text{ if, and only if} \tag{G0a}
$$

$$
f_{g_0/l}(l^2 U) = l^n f_{g_0}(U)\,, \qquad \text{at some } g_0 \text{ in the carrier of interest, and} \tag{G0i}
$$

$$
\frac{d}{dg} f_{g/l}(l^2 U) = \frac{d}{dg} l^n f_g(U)\Big|_{\frac{d}{dg}f(g/l)=\frac{1}{l}\frac{d}{d(g/l)}f(g/l)=\frac{1}{l}\dot{f}(g/l)}
$$

$$
\Longleftrightarrow \dot{f}_{g/l}(l^2 U) = l^{n+1} \dot{f}_g(U)\,, \qquad \text{on the whole carrier.} \tag{G0d}
$$

We have to verify these conditions for the case of an ODE, i.e. not for given explicit expressions, but starting from initial conditions at $g_0 = 0$, and we need to show that the following relations hold for all $g$ in the carrier, grouping the equations such that the role of the initial condition

is on the left, and the resulting relations implied by these conditions on the right:

$$\Sigma_{g/l}(l^2 U) = l \Sigma_g(U), \qquad \text{(S0)} \qquad\qquad \ddot{\Sigma}_{g/l}(l^2 U) = l^3 \ddot{\Sigma}_g(U), \qquad \text{(S2)}$$

$$\dot{\Sigma}_{g/l}(l^2 U) = l^2 \dot{\Sigma}_g(U), \qquad \text{(S1)} \qquad\qquad \dot{V}_{g/l}(l^2 U) = l^3 \dot{V}_g(U). \qquad \text{(V1)}$$

$$V_{g/l}(l^2 U) = l^2 V_g(U), \qquad \text{(V0)}$$

To prove G0d for this set on the whole carrier, we argue similar to [21] as follows: Given that S0, S1 and V0 hold at some $g$, we will show that this implies also S2 and V1 to hold *at that point* $g$. In return, by means of S2 and V1 at that point $g$, an infinitesimal step by means of the closed set of differential equations then ensures that S0, S1 and V0 remain valid at $g + \delta g$, and with this also S2 and V1 remain valid at $g + \delta g$, and thus by integration/repetition on the whole carrier. This completes the argument and is a differential version of the discrete induction proof in [21].[8]

### B.3.1 Scaling of standard propagators

Core quantities on the rhs of the flow equation are the propagators, standard and "derived" ones. We first assume that the conditions S0, S1 and V0 hold at some $g$ and look at the behaviour of S2 and V1 under this assumption. This also requires an analysis of the $Z$-factors involved. By means of the definition we have [24]

$$G_g(U) = \frac{g}{i\omega - \xi_0 - g\Sigma_g(U)},$$

$$G_{g/l}(l^2 U) = \frac{g/l}{i\omega - \xi_0 - \frac{g}{l}\Sigma_{g/l}(l^2 U)} \qquad \Big|_{\text{use assumption S0}}$$

$$= \frac{1}{l} \frac{g}{i\omega - \xi_0 - g\Sigma_g(U)}$$

$$= \frac{1}{l} G_g(U).$$

The approximation of the full scale-dependent propagator by inclusion of a quasi-particle weight reads [24]

$$G_g(U) \approx \frac{g Z_g}{i\omega - \xi_0},$$

and for $Z$ we have

$$Z_g(U) = Z(g\Sigma_g(U)) = \frac{1}{1 - \partial_\omega \text{Re}(g\Sigma_g(U)|_{\omega=0, \mathbf{k}=\mathbf{k}_F}}$$

$$Z_{g/l}(l^2 U) = Z(\frac{g}{l}\Sigma_{g/l}(l^2 U)) = \frac{1}{1 - \partial_\omega \text{Re}(\frac{g}{l}\Sigma_{g/l}(l^2 U)|_{\omega=0, \mathbf{k}=\mathbf{k}_F}} \qquad \Big|_{\text{use assumption S0}}$$

$$= \frac{1}{1 - \partial_\omega \text{Re}(g\Sigma_g(U)|_{\omega=0, \mathbf{k}=\mathbf{k}_F}}$$

$$= (l^0) Z_g(U),$$

---

[8]This is actually quite a strong implication: If the conditions hold at one point, they hold everywhere. In that sense they are as firm as the uniqueness property for an ODE, i.e. when two functions have the same value at some point and fulfil the same ODE, they are identical. But here the case is more general.

and thus

$$G_{g/l}(l^2U) \approx \frac{g}{l} \frac{Z_{g/l}(l^2U)}{i\omega - \xi_0}$$

$$= \frac{1}{l} G_g(U).$$

The scaling property thus remains valid. Also, if we omit the self-energy feedback altogether, this property is trivial by definition.

### B.3.2 Scaling of derived propagators

The single-scale propagator is defined as

$$S_g(U) = \frac{i\omega - \xi_0}{(i\omega - \xi_0 - g\Sigma_g(U))^2}$$

$$S_{g/l}(l^2U) = \frac{i\omega - \xi_0}{(i\omega - \xi_0 - \frac{g}{l}\Sigma_{g/l}(l^2U))^2} \qquad \Big|_{\text{use assumption S0}}$$

$$= \frac{i\omega - \xi_0}{(i\omega - \xi_0 - g\Sigma_g(U))^2}$$

$$= (l^0)S_g(U).$$

In the quasi-particle flow it is approximated as

$$S_g(U) \approx \frac{Z_g^2}{i\omega - \xi_0}.$$

Thus, by virtue of the above scaling of $Z$ the scaling for $S_g$ also remains valid.
Including the Katanin correction leads to the use of a derived full propagator:

$$\dot{G}_g(U) = \frac{1}{i\omega - \xi_0 - g\Sigma_g(U)} + g\frac{\Sigma_g(U) + g\dot{\Sigma}_g(U)}{(i\omega - \xi_0 - g\Sigma_g(U))^2},$$

$$\dot{G}_{g/l}(l^2U) = \frac{1}{i\omega - \xi_0 - \frac{g}{l}\Sigma_{g/l}(l^2U)} + \frac{g}{l}\frac{\Sigma_{g/l}(l^2U) + \frac{g}{l}\dot{\Sigma}_{g/l}(l^2U)}{(i\omega - \xi_0 - \frac{g}{l}\Sigma_{g/l}(l^2U))^2} \Big|_{\text{use assumptions S0 and S1}}$$

$$= \frac{1}{i\omega - \xi_0 - g\Sigma_g(U)} + g\frac{\Sigma_g(U) + g\dot{\Sigma}_g(U)}{(i\omega - \xi_0 - g\Sigma_g(U))^2}$$

$$= \dot{G}_g(U).$$

In the quasi-particle flow the full derivative is obtained as [24]

$$\dot{G}_g(U) = K_g(U)S_g(U), \qquad \text{with} \qquad K_g(U) = \left(1 + g^2\partial_\omega \text{Re}\left(\dot{\Sigma}_g(U)\right)\big|_{\omega=0,\mathbf{k}=\mathbf{k}_F}\right).$$

Since

$$K_{g/l}(l^2U) = \left(1 + \left(\frac{g}{l}\right)^2\partial_\omega \text{Re}\left(\dot{\Sigma}_{g/l}(l^2U)\right)\big|_{\omega=0,\mathbf{k}=\mathbf{k}_F}\right)\Big|_{\text{use assumption S1}}$$

$$= \left(1 + \left(\frac{g}{l}\right)^2\partial_\omega \text{Re}\left(l^2\dot{\Sigma}_g(U)\right)\big|_{\omega=0,\mathbf{k}=\mathbf{k}_F}\right)$$

$$= \left(1 + g^2\partial_\omega \text{Re}\left(\dot{\Sigma}_g(U)\right)\big|_{\omega=0,\mathbf{k}=\mathbf{k}_F}\right)$$

$$= K_g(U),$$

we have the same scaling for $K_g$ as for $S_g$ and thus for their product. Also, when neglecting self-energy feedback the same scaling applies trivially.

To conclude and generalise, propagators $P_g(U)$ and derived propagators $dP_g(U)$ scale like

$$P_{g/l}(l^2 U) = l^{-1} P_g(U), \tag{P0}$$

$$dP_{g/l}(l^2 U) = l^0 dP_g(U), \tag{P1}$$

where $P$ and $dP$ represent whichever type of dressed, non-dressed or approximately dressed versions we use.

### B.3.3 Scaling consistency for the flow of the effective interaction

We proceed to analyse the behaviour of condition V1. The flow equation consists of diagrams of the type

$$\dot{V}_g(U) = \left(V_g(U)\right)^2 P_g(U) dP_g(U),$$

$$\dot{V}_{g/l}(l^2 U) = \left(V_{g/l}(l^2 U)\right)^2 P_{g/l}(l^2 U) dP_{g/l}(l^2 U) \Big|_{\text{use assumptions/conditions V0,P0,P1}}$$

$$= \left(l^2 V_g(U)\right)^2 l^{-1} P_g(U) l^0 dP_g(U)$$

$$= l^3 \dot{V}_g(U),$$

fulfilling the condition V1.

### B.3.4 Scaling consistency for the flow of the self-energy

The flow of the self-energy is computed based on the second derivative. It has to contributions, an effective one-loop part and a two-loop part:

$$\ddot{\Sigma}_g(U) = \ddot{\Sigma}_g^{1l}(U) + \ddot{\Sigma}_g^{2l}(U),$$

The two-loop part $\ddot{\Sigma}_g^{2l}(U)$ is of the sunset type and has the structure [24]

$$\ddot{\Sigma}_g^{2l}(U) = \left(V_g(U)\right)^2 P_g(U) dP_g(U) dP_g(U),$$

$$\ddot{\Sigma}_{g/l}^{2l}(l^2 U) = \left(V_{g/l}(l^2 U)\right)^2 P_{g/l}(l^2 U) dP_{g/l}(l^2 U) dP_{g/l}(l^2 U) \qquad \Big|_{\text{use V0,P0,P1}}$$

$$= \left(l^2 V_g(U)\right)^2 l^{-1} P_g(U) l^0 dP_g(U) dP_g(U)$$

$$= l^3 \ddot{\Sigma}_g^{2l}(U),$$

fulfilling the condition S2.

The one-loop part $\ddot{\Sigma}_g^{1l}(U)$ is an approximate term involving $\dot{\Sigma}$ and reads [24]

$$\ddot{\Sigma}_g^{1l}(U) = \bar{A}_g(U) \dot{\Sigma}_g(U).$$

Here,

$$A_g(U) = 2 \frac{\dot{Z}_g(U)}{Z_g(U)}, \qquad \text{and} \qquad \bar{A} = \text{average}(A)\big|_{\text{Fermi surface}},$$

To check the scaling of this expression we also need to refer to the expression that results for $\dot{Z}_g$ [24], omitting the explicit restriction for the $\omega$-derivative for readability:

$$\dot{Z}_g(U) = Z_g^2(U)\left\{\partial_\omega \text{Re}\left(\Sigma_g(U)\right) + \partial_\omega \text{Re}\left(g\dot{\Sigma}_g(U)\right)\right\},$$

$$\dot{Z}_{g/l}(l^2 U) = Z_{g/l}^2(l^2 U)\left\{\partial_\omega \text{Re}\left(\Sigma_{g/l}(l^2 U)\right) + \partial_\omega \text{Re}\left(\frac{g}{l}\dot{\Sigma}_{g/l}(l^2 U)\right)\right\}$$

$$= Z_g^2(U)\left\{\partial_\omega \text{Re}\left(l\Sigma_g(U)\right) + \partial_\omega \text{Re}\left(gl\dot{\Sigma}_g(U)\right)\right\}$$

$$= lZ_g^2(U)\left\{\partial_\omega \text{Re}\left(\Sigma_g(U)\right) + \partial_\omega \text{Re}\left(g\dot{\Sigma}_g(U)\right)\right\}$$

$$= l\dot{Z}_g(U), \tag{Z1f}$$

where we used S1 and the previous scaling properties and conditions from above. We then get

$$A_{g/l}(l^2 U) = 2\frac{\dot{Z}_{g/l}(l^2 U)}{Z_{g/l}(l^2 U)} = 2l\frac{\dot{Z}_g(U)}{Z_g(U)} = lA_g(U)$$

$$\implies \bar{A}_{g/l}(l^2 U) = l\bar{A}_g(U), \tag{B.1}$$

with the scaling carrying over to the average $\bar{A}$. We can now evaluate the scaling for the one-loop contribution:

$$\ddot{\Sigma}_{g/l}^{1l}(l^2 U) = \bar{A}_{g/l}(l^2 U)\dot{\Sigma}_{g/l}(l^2 U)$$

$$= l\bar{A}_g(U)l^2\dot{\Sigma}_g(U)$$

$$= l^3\ddot{\Sigma}_g^{1l}(U), \tag{B.2}$$

in line with S2 and the scaling of the two-loop part.

At this point, we have verified that the flow equations consistently conserve the scaling properties, the remaining condition being their fulfilment at a specific value of $g$. The natural choice for this are the initial conditions of the flow at $g_0 = 0$.

### B.3.5 Scaling property for initial conditions

Let us check the initial conditions:

Condition S0 is trivial since the self-energy starts at zero:

$$\Sigma_{(g/l)=0}(l^2 U) = 0,$$

$$l\Sigma_{g=0}(U) = 0.$$

Condition V0 is set by the very definition of the bare interaction and actually *determines* the exponent of the internal scaling with respect to the dependence of the functions on the bare interaction:

$$V_{(g/l)=0}(l^2 U) = (l^2 U),$$

$$l^2 V_{g=0}(U) = l^2(U).$$

What remains to be checked is a slightly more tricky part, concerning condition **S1**. In fact, this term does not vanish at $g = 0$. Rather, it is a Hartree-type contribution. Without resorting to the second derivative the right-hand side for flow of the self-energy at $g = 0$ reads

$$\dot{\Sigma}_{(g/l)=0}(l^2 U) = (l^2 U) \circ G^0 = \frac{(l^2 U)n_0}{2},$$

$$l^2 \dot{\Sigma}_{g=0}(U) = l^2(U \circ G^0) = l^2 \frac{(U)n_0}{2}.$$

While this verifies S1, we recall that we compute the flow of the *imaginary part* only. The initial Hartree-contribution to $\dot{\Sigma}$ is however purely real, and thus for our purposes the condition S1 is also trivially fulfilled since it vanishes for the imaginary part of the self-energy at $g = (g/l) = 0$. However, the fact that we keep the Fermi surface fixed and neglect the Hartree contribution calls for some more comments, which we shall offer below in section C. What really happens at the conceptual level is an implicit change of the chemical potential during the flow, which keeps the Fermi surface at its non-interacting location.

For the sake of completeness of the equations at the initial value, we can double-check the validity of V1 and S2 at $g = g_0 = 0$. Condition **V1** is again trivial since it vanishes at $g = 0$ due to a factor of $g$ in one of the internal propagators on the rhs:

$$\dot{V}_{(g/l)=0}(l^2 U) = (l^2 U)^2 P_{(g/l)=0}(l^2 U) dP_{g/l}(l^2 U) = 0,$$

$$\dot{V}_{g=0}(U) = U^2 P_{g=0}(U) dP_g(U) = 0.$$

Also **S2** is fulfilled at $g = 0$. For the two-loop part this is due to the vanishing of one of the internal propagators:

$$\ddot{\Sigma}^{2l}_{(g/l)=0}(l^2 U) = (l^2 U)^2 P_{(g/l)=0} dP_{g/l} dP_{g/l} = 0,$$

$$\ddot{\Sigma}^{2l}_{g=0}(U) = U^2 P_{g=0} dP_g dP_g = 0.$$

The one-loop part vanishes due to the vanishing of $\dot{Z}$ according to Z1f . With $Z_{g=(g/l)=0} = 1$ this translates to

$$\dot{Z}_{(g/l)=0}(l^2 U) = \partial_\omega \text{Re}\left(\Sigma_{(g/l)=0}(l^2 U)\right) + \partial_\omega \text{Re}\left(\frac{g}{l}\dot{\Sigma}_{(g/l)=0}(l^2 U)\right) = 0,$$

$$\dot{Z}_{g=0}(U) = \partial_\omega \text{Re}\left(\Sigma_{g=0}(U)\right) + \partial_\omega \text{Re}\left(g\dot{\Sigma}_{g=0}(U)\right) = 0.$$

The first terms vanishes, since the self-energy itself vanishes, and the second due to the factor $g = (g/l) = 0.$[9]

### B.3.6 Scaling summary

We have shown that the conditions that are required for the scaling property of the flow equations are obeyed by the specific implementation and approximations upon which we have based the numerical treatment. The scaling condition is mandatory for the interpretation of the flow as a gradual increase of the bare interaction. The fact that all derivatives vanish at the initial point is somewhat more demanding for the algorithm that solves the ODE, in particular when using an explicit forward scheme as done here, and it is important to choose a small enough initial step size. We verified the scaling property for some test cases explicitly, since it also provides an important check for the numerical implementation.

---

[9]It is also sufficient that both terms are frequency-independent and thus the frequency-derivatives vanish.

# C   Hartree-shift and fixating the Fermi surface

In perturbative, diagrammatic treatments, the lowest order contribution to the self-energy of a many-body system of interacting electrons is given by the Hartree and Fock terms, computed in the non-self-consistent version, and for the local density-density interaction in the Hubbard model the Fock term vanishes.[10]

In fRG treatments, the Hartree term is often eliminated from the formalism, the argument for instance being that it is merely a shift in the chemical potential. If we neglect self-energy contributions on the rhs of the flow equation altogether, the character of the resulting fRG approximation is analogous to bare perturbation theory, in which the self-energy is computed using non-interacting propagators and vertices, and then plugged into the Dyson equation, followed by a one-step shift of the chemical potential if the density shall be kept constant.

Yet, it is one of the very virtues of fRG equations that they are *exact* in their fundamental and complete form, and leaving out *any* terms ad hoc in principle spoils that very property. Still, this is often done in practical applications, let alone for reasons of feasibility, not only by neglecting self-energy effects, but also by truncating the hierarchy of equations, simplifying the parametrisation, etc. With respect to the Hartree shift, in the interaction flow the matter becomes more transparent: This flow smoothly connects *solutions* for different bare interactions, and the flow equation *does* include the Hartree term. Obviously, we cannot appropriately adjust the chemical potential for *all* these bare interactions using only *one* simple shift in the chemical potential. We can - and do - of course omit the self-energy in the propagators altogether in many fRG applications, but that really is not a very controlled mathematical step, and it would be nice to justify or at least understand it a little better. In particular, when we *do* include self-energy effects in propagators on the rhs of the flow equation, we should worry first and foremost about the lowest order contribution to this. While we will not be able to clarify this aspect completely, we will offer some views to better understand some of the approximations that are involved when we keep the Fermi surface fixed by keeping the bare dispersion in the propagators, while "only" adding effects from the frequency-dependence of the self-energy via a quasi-particle weight.

In what follows, we will walk through the probably simplest possible case of a mean-field approximation in a two-dimensional Fermi system, namely the self-consistent Hartree approximation in the normal phase, considering the total density as the "order parameter".[11] The main purpose of the exercise will then be an elementary comparison to approximations that stem from the 1-PI fRG equation, related to the connection of the Katanin replacement [30] to self-consistent mean-field-type solutions, guided by the route in [32, 66, 67].

## C.1   Toy model

We resort to a very simple model at $T = 0$ with a local interaction. We use a simple quadratic dispersion, as e.g. taken from the low-density expansion of a lattice dispersion, and write

$$\epsilon(\mathbf{k}) = t\,(k_x^2 + k_y^2) = t\,|\mathbf{k}|^2\,. \tag{C.1}$$

Denoting by $n(\epsilon)$ the density for all modes with energies up to $\epsilon$, and $|\mathbf{k}(\epsilon)|$ the radius of the

---

[10]More precisely speaking, the Fock term vanishes upon fixing the quantisation axis and choosing to decouple the degrees of freedom with respect to the density, see F. Lechermann in [65].

[11]Note that this excludes *a priori* other kinds of sectors in which instabilities may or may not arise through mean-filed solutions, such as e.g. Stoner magnetism, superconductivity or charge-density waves.

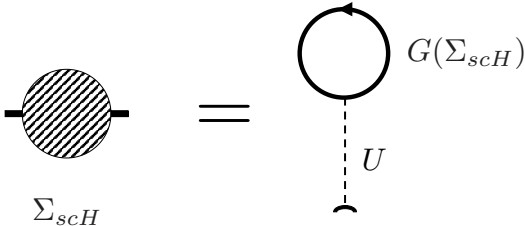

Figure 22: Diagrammatic representation of the self-consistent Hartree approximation to the self-energy.

circle enclosing them, we have

$$n(\epsilon) \propto 2\pi \, |\mathbf{k}(\epsilon)|^2 = 2D\epsilon \, . \tag{C.2}$$

Here, we implicitly define the total density of states (DOS) of the non-interacting system as $2D$, the factor 2 stemming from the spin degree of freedom. We note that D is constant, i.e. independent of $\epsilon$.

In the grand-canonical framework, the density of the free system is controlled by the chemical potential $\mu$, which also defines the free Fermi surface via $\epsilon(\mathbf{k}_F) = \mu$. At constant $D$ and at $T = 0$, by Eq. C.2 it is thus simply given as

$$n(\mu) = 2\mu D \, . \tag{C.3}$$

We further assume that the bare two-particle interaction is given by a simple local term $U$.

## C.2 Self-consistent Hartree approximation

The self-consistent Hartree approximation (scH) consists in an implicit equation for the self-energy $\Sigma_{scH}$ at first order in the bare interaction from a tad-pole diagram, as sketched in Fig. 22.
Upon fixing the quantisation axis, the bare interaction couples spin-up electrons only with spin-down electrons and vice versa, and the self-consistency equation reads

$$\Sigma_{scH} = \frac{1}{2} U \, n(\mu, \Sigma_{scH}) \, . \tag{C.4}$$

By means of the Dyson equation we have[12]

$$G_0^{-1}(i\omega, \mathbf{k}) = i\omega_n - \epsilon_{\mathbf{k}} + \mu \, ,$$

$$G^{-1}(i\omega, \mathbf{k}) = G_0^{-1} - \Sigma(i\omega, \mathbf{k}) \, .$$

Since the bare interaction is chosen as a local density-density-coupling $U$, $\Sigma_{scH}$ is a simple number, independent of frequency and momentum. It depends only on $\mu$ and $U$ and is often referred to as a shift in the chemical potential. Further more, in the resulting description of

---

[12]This is already a non-trivial aspect, since the Dyson equation as such constitutes a resummation to infinite order in $U$ and is thus non-perturbative in nature. It is a *choice* to use it and to apply perturbative approximations to the *self-energy*, rather than computing corrections to the propagator directly.

the interacting system the density $n(\mu, \Sigma_{scH})$ depends on $\mu$ and $\Sigma_{scH}$ only via the difference $\mu - \Sigma_{scH}$, with propagators remaining structurally as they are in the free system. We can thus use the very same constant DOS as in the non-interacting system to compute the density $n(\mu, \Sigma_{scH})$ and can explicitly write the self-consistency equation *and* its solution:

$$\Sigma_{scH} = U \frac{n}{2}(\mu, \Sigma_{scH}) = U D (\mu - \Sigma_{scH}) \tag{C.5}$$

$$\implies \qquad \Sigma_{scH} = \mu \frac{UD}{1+UD}$$

$$\implies \qquad \tilde{\mu} := \mu - \Sigma_{scH} = \mu \frac{1}{1+UD} \,.$$

Here, we chose $\tilde{\mu}$ to define an effective chemical potential. We can make some simple plausibility checks:

- $\mu \to 0$, at $U$ fixed $\implies \Sigma_{scH} \to 0$ and $\tilde{\mu} \to 0$: When the free system becomes empty, so remains the interacting one. "No particles, no effect".

- $U \to 0$, at $\mu$ fixed $\implies \Sigma_{scH} \to 0$: When switching off the interaction, the self-energy vanishes. "No interaction, no change".

- $U \to +\infty$ at $\mu$ fixed $\implies \Sigma_{scH} \to \mu^-$, $\tilde{\mu} \to 0^+$: Cranking up a repulsive interaction at fixed chemical potential introduces higher costs for adding an electron to the system. Thus, the filling in the interacting system is reduced. But it cannot be reduced below zero filling, and that is indeed the asymptotics here.

- For a negative interaction, the opposite is the case: Particles get more and more "sucked in" when cranking up an attraction, and thus $\tilde{\mu}$ and with it the filling increases. There is a simple view on this, when we treat the interaction energy via this mean-field approximation:[13] At $1 = UD$ we reach the point where the gain in potential energy upon adding a particle exceeds the cost in kinetic energy: $\delta E_{kin} = \mu \delta n$, $\delta E_{int} = (Un/2)\delta n = \mu UD\delta n$, and thus $\delta E = \mu(1 + UD)\delta n$. For $UD \to -1$ from above the cost off adding a particle goes to zero and the situation becomes instable. Of course, real systems are usually defined by a given density or at least a limit thereof, and we use the grand canonical description only for practical purposes, to later invert the relation $n(\mu)$ after all calculations are done. E.g. for the Hubbard model this means that at a certain value of $U$ the filling will reach its upper limit for the case of an attractive interaction. Further increasing the interaction strength then pushes the band below the (external) chemical potential and there is no one-to-one mapping between $\mu$ and $n$ anymore, at least not without further arguments or extending the situation to finite temperature.

We finally note that we can alternatively write the self-consistency equation in terms of the (interacting) density $n_{scH}$:

$$2D(\mu - \Sigma_{scH}) = n_{scH} = 2D\left(\mu - U\frac{n_{scH}}{2}\right)$$

$$\implies n_{scH} = \frac{2D\mu}{1+UD} = \frac{n_{free}}{1+UD} \,. \tag{C.6}$$

This is related to Kanamori screening in the context of the (in)stability of the paramagnetic state towards ferromagnetism [68].

---

[13]This is analogous to a familiar criterion for Slater ferromagnetism, but simpler.

## C.3 Contact with fRG and interaction flow

The above calculations are simple and of one-step character. The issue with which we are concerned appears when we try to approach the self-consistent solution *continuously* in some parameter, e.g. $U$, rather that by a single-step procedure at *fixed* parameters. Such a continuous approach is inherent in fRG calculations, one of the core issues being the movement and/or deformation of the Fermi surface, i.e. the change of the manifold of infrared singularities in the propagator. The above sketched scH approximation has prepared us to isolate this very effect in a most simplistic manner.

We recall the definition of the scale-dependent propagator in the interaction flow version of the fRG [11]:

$$G_{0,g} = \frac{g}{i\omega_n - \epsilon_{\mathbf{k}} + \mu},$$

$$G_g^{-1} = G_{0,g}^{-1} - \Sigma_g = \frac{i\omega_n - \epsilon_{\mathbf{k}} + \mu}{g} - \Sigma_g,$$

$$i.e. \qquad G_g = \frac{g}{i\omega_n - \epsilon_{\mathbf{k}} + \mu - g\Sigma_g}.$$

Here, as in the main part of this work, $g$ is the flow parameter and amounts to the homogeneous and continuous activation of all modes. In the 1-PI scheme which we use, we can interpret this scaling as a continuous increase of the bare interaction [11]. As outlined above, the corresponding scaling property of the interaction flow with given bare interaction $U$ reads

$$g\Sigma_g(U) = \Sigma_{g=1}(g^2 U) = \Sigma_{final}(g^2 U),$$

and as usual allows to read off the final solution for a bare interaction of the strength $g^2 U$ by taking the value of the self-energy at that value of $g$ and multiplying it by $g$.

An important step in the evolution of fRG methods was the observation that self-consistent, "mean-field-exact" approximations can be reproduced in the 1-PI fRG by using a subset of RPA contributions to the flow of the effective interaction *and* applying the Katanin replacement (RPA+Katanin) [30,32,67]. The resulting equations of this procedure are summarised graphically in Figure 23. To follow this very mechanism for our simplistic case, we start from the known solution above and will then make contact with the fRG path. Similar to the results in the main part, we will also contrast the self-consistent case with two other cases, when the equations are truncated after first order in $U$, namely the standard fRG case as well as the again simpler case where the feedback of the self-energy is completely neglected on the rhs of the flow equation, see Figure 24.

**Building the bridge from the self-consistency equation**

We operate on the self-consistency equation (C.5) directly and suitably insert the scaling parameter $g$, to extract an expression which permits a comparison to the fRG route. Defining (!) $\Sigma_g$ via $g\Sigma_g := \Sigma(g^2 U)$ and omitting the label *scH* for brevity, the $g$-dependent self-consistency

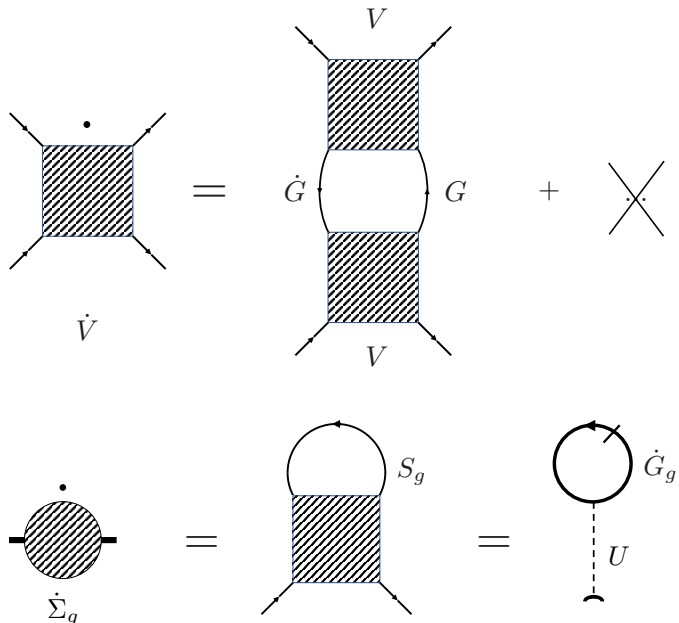

Figure 23: The RPA+Katanin subset of fRG equations for the effective interaction reproduces the self-consistent Hartree solution for the self-energy, c.f. Figure 2 in [32].

equation reads

$$\Sigma(g^2 U) = g^2 U\, n(\mu, \Sigma(g^2 U)) = g^2 UD\,(\mu - \Sigma(g^2 U))$$

$$\implies \Sigma_g = gUD\,(\mu - g\Sigma_g)$$

$$\implies \dot\Sigma_g := \frac{d}{dg}\Sigma_g = UD\mu - 2gUD\Sigma_g - g^2 UD\dot\Sigma_g$$

$$\implies (1 + g^2 UD)\dot\Sigma_g = UD\mu - 2gUD\Sigma_g. \tag{C.7}$$

While this expression will be sufficient to check the matching to the fRG case, we can go a little further if we wish:

$$\implies \dot\Sigma_g = \frac{UD\mu - 2gUD\Sigma_g}{1 + g^2 UD}\, \Bigg|_{g\Sigma_g = \Sigma(g^2 U) = g^2 UD/(1 + g^2 UD)}$$

$$\implies \dot\Sigma_g = \frac{\mu\Sigma_g}{g} - 2\Sigma_g^2.$$

**Building the bridge from the "mean-field-exact" fRG equation** Coming from the fRG side, the self-consistent Hartree solution is obtained by replacing the single-scale propagator $S_g$ in the one-loop equation for the effective interaction by the total derivative $\dot G_g$ of the full

$$X_g = \begin{cases} \dot{G}_g & \text{RPA+Katanin} \\ S_g & \text{standard fRG equation} \\ \dot{G}_{0,g} & \Sigma\text{-feedback neglected} \end{cases}$$

Figure 24: Three variants of the flow equation for the self-energy that are compared for the toy model. While the standard fRG equation and the case without feedback stem from a hard truncation of the hierarchy after first order in U, the equation that reproduces the self-consistent Hartree solutions requires the RPA+Katanin choice for the flow of the effective interaction.

propagator at scale $G$, as it results from the RPA+Katanin subset, as mentioned above [32]. This derivative reads

$$\dot{G}_g = \frac{d}{dg} \frac{g}{i\omega_n - \epsilon_{\mathbf{k}} + \mu - g\Sigma_g}$$

$$= \frac{1}{i\omega_n - \epsilon_{\mathbf{k}} + \mu - g\Sigma_g} + \frac{g(\Sigma_g + g\dot{\Sigma}_g)}{(i\omega_n - \epsilon_{\mathbf{k}} + \mu - g\Sigma_g)^2}.$$

The one-loop flow equations for the self-energy then yields, by means of the usual analytic treatment of the Matsubara sums,

$$\dot{\Sigma}_g = UD(\mu - g\Sigma_g) + gU(\Sigma_g + g\dot{\Sigma}_g)) \int d^2k f'(\epsilon_{\mathbf{k}} - \mu + g\Sigma_g)$$

$$= UD(\mu - g\Sigma_g) + gU(\Sigma_g + g\dot{\Sigma}_g)) \int d\epsilon D(\epsilon)(-\delta(\epsilon_{\mathbf{k}} - \mu + g\Sigma_g))$$

$$= UD(\mu - g\Sigma_g) - gUD(\Sigma_g + g\dot{\Sigma}_g))$$

$$\implies (1 + g^2 UD)\dot{\Sigma}_g = UD\mu - 2gUD\Sigma_g, \tag{C.8}$$

and equation (C.8) coincides with equation (C.7). Here, we have directly worked with the one-loop equation for the self-energy that results from the completion of $S$ to $\dot{G}$ *before* solving the fRG equation [32]. This equation is nothing else but the *derivative* of the self-consistency equation, but in this step we evaluated it along the fRG route. With identical initial conditions, we thus arrive at the same result that we got when taking the derivative of the self-consistency equation directly. This may seem of little surprise, since we have been massaging the very same approximation, only from different angles. Yet, it is the very *specific* choice of the RPA+Katanin subset of the 1-PI fRG equations that ensures this. This choice does not emerge from the fRG formalism in an obvious or natural manner, but was identified step-wise, or rather discovered. In particular, a technically trivial consequence of this is that the "normal" rhs of the truncated 1-PI equation does *not* reproduce the self-consistent solution. Thus, we shall now compare the result we have obtained so far to the flow equation we get *without* feeding the Katanin-corrected RPA-ladder back into the equation of the self-energy, i.e. we will truly truncate the

hierarchy at first order in the effective interaction and use the single-scale propagator $S_g$ to repeat the calculation.

The single-scale propagator is given as

$$S_g = -G_g \dot{G}_{0,g}^{-1} G_g$$

$$= \frac{g}{i\omega_n - \epsilon_{\mathbf{k}} + \mu - g\Sigma_g} \frac{i\omega_n - \epsilon_{\mathbf{k}} + \mu}{g^2} \frac{g}{i\omega_n - \epsilon_{\mathbf{k}} + \mu - g\Sigma_g}$$

$$= \frac{1}{i\omega_n - \epsilon_{\mathbf{k}} + \mu - g\Sigma_g}(i\omega_n - \epsilon_{\mathbf{k}} + \mu - g\Sigma_g + g\Sigma_g)\frac{1}{i\omega_n - \epsilon_{\mathbf{k}} + \mu - g\Sigma_g}$$

$$= \frac{1}{i\omega_n - \epsilon_{\mathbf{k}} + \mu - g\Sigma_g} + \frac{g\Sigma_g}{(i\omega_n - \epsilon_{\mathbf{k}} + \mu - g\Sigma_g)^2}.$$

This is the usual result, i.e. $S_g$ differs from $\dot{G}_g$ by the term involving $\dot{\Sigma}_g$. For the calculation as such we can simply read off the result from the previous calculation that led to equation (C.8) and omit the $\dot{\Sigma}_g$ term to arrive at

$$\dot{\Sigma}_g = UD\mu - 2gUD\Sigma_g$$

$$= UD(\mu - 2g\Sigma_g). \tag{C.9}$$

This time, the flow equation *can not* be simplified further by reinserting the explicit self-consistent solution, since this equation does not reproduce it. Also, we need to take some care: If $\mu - g\Sigma_g$ was to change sign, the integral over the derivative of the Fermi function $f'$ would not run over the peak of the delta function anymore and the second term would vanish. In fact, that will not happen since at an earlier point a sign change takes place when $(\mu - 2g\Sigma_g) = 0$, leading to a maximum, followed by a decrease. This, however, is unphysical, since it means that further increasing the repulsion leads to a re-increase of the density. Beyond the maximum, the curve will asymptotically saturate when $(\mu - 2gUD\Sigma_g) \to 0$, due to the decrease of $\Sigma_g$.

In many fRG calculations the self-energy is neglected completely on the right-hand side of the flow equation, which here yields

$$\dot{\Sigma}_g = UD\mu.$$

This of course is nothing but the flow version of the *non-self-consistent* Hartree approximation, yielding $g\Sigma_g = g^2 UD\mu.$[14]

The genuine, standard fRG equation for the self-energy, when the exact hierarchy is truncated after first order in the effective interaction without any further modifications, thus seems to fall in between the non-self-consistent and the self-consistent Hartree approximation. Also, while there are explicit expressions for the self-energy in the latter two, this is not so simple for the standard fRG case.

---

[14]Note that the property of the interaction flow regarding the equivalence of scaling the interaction and scaling the flow parameter is valid for all these cases. This is "easy" to verify, c.f. [21], but in general by no means guaranteed to be the case. When working with counter terms, or additional terms that are fixed at the end of the flow at $g = 1$, this may be violated. Then, we have to be careful when interpreting the flow.

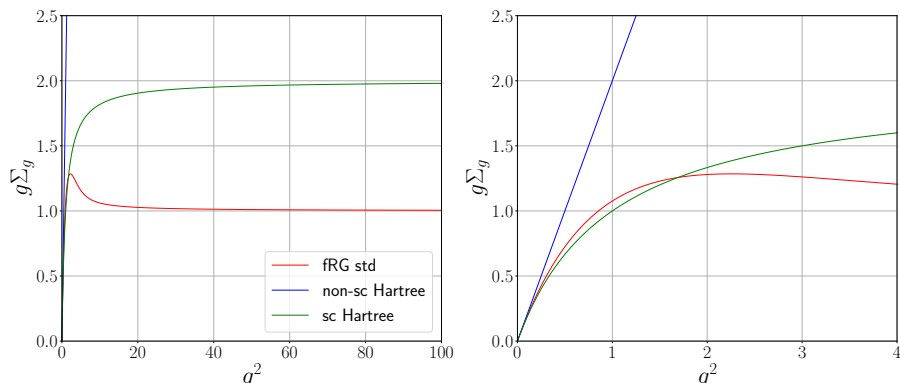

Figure 25: Comparison of the three versions of the self-energy flow $g\Sigma(g)$ for $UD = 1$ and $\mu = 2$ as a function of $g^2$.

As an illustrating example, it is is instructive to perform a numerical treatment of the three cases we now have at hand. The result of this is shown in Figure 25, for the choice $UD = 1$ and $\mu = 2$, where we plot $g\Sigma_g$ as a function of $g^2$, i.e. the final solution for the self-energy as a function of the bare coupling. We note:

- All curves start with the same value and slope, as they should.

- The non-self-consistent straight line is familiar. When it reaches $g\Sigma_g = \mu$ we know that the resulting interacting system is empty and beyond that point the curve is meaningless.

- The self-consistent, i.e. RPA+Katanin, case is in line with the checks above: Increasing the interaction squeezes particles out of the system, and the empty state is reached asymptotically.

- The standard fRG case, however, is particular in two aspects:
  i) The maximum of $\Sigma_g$ mentioned above translates into a maximum of $g\Sigma_g$ at some point $g^2 = U \approx 2$, to then fall off again.
  ii) $g\Sigma_g$ reaches an asymptotic value that is half of the self-consistent one.
  Both aspects can easily be understood from the equation, but are physically not sound: The system should become empty for large interaction. In the non-self-consistent case this happens way too early, i.e. for much too small interaction, but it *happens*. In the self-consistent case it happens asymptotically, for large $UD$. But for the standard fRG case it does *not happen at all*. Instead, half of the original particles stay in the system. The maximum is even more unphysical, since increasing a repulsion can certainly not lead to a (re-)increase of the density. Clearly, both effects are unphysical and to our understanding likely related to the violation of Ward identities, the reduction of which was the very driving force leading to the suggestion of the Katanin replacement [30, 69].

This very simple example provides an explicit and instructive view on how the standard fRG equation for the self-energy relates to the self-consistent one. As stated before, we know from general arguments that we can obtain a self-consistent solution for the self-energy by *restricting* the flow of the effective interaction to the *subset* of the RPA channel *and* at the same time *adding* contributions from higher orders of the hierarchy via the Katanin replacement. The familiar solution of this RPA being $V_g = U/(1 - U\Pi_{ph})$ yields $V_g = U/(1 + g^2 UD\mu)$. Indeed, that is

what we find when comparing equation C.8 and C.9:

$$(1 + g^2 UD)\dot{\Sigma}_g = UD\mu - 2g UD\Sigma_g$$

$$\implies \dot{\Sigma}_g = \underbrace{\frac{U}{(1 + g^2 UD)}}_{V_{g,RPA}} \underbrace{\left( D\mu - 2g D\Sigma_g \right)}_{\text{Standard loop over } S_g} = V_g \circ S_g \,.$$

The fact that the convolution $V \circ \Sigma$ factorises into a simple product is of course due to the restriction to the single RPA channel.

A next step towards a more fair comparison between the standard fRG equations and the Katanin replacement would be the inclusion of second order terms in the effective interaction and keeping the single-scale propagator within the RPA calculation. This is however beyond the scope of the conceptual illustration we try to give here. However, with increasing truncation level we expect that deviations become smaller. This can only be checked numerically, since the usage of $S_g$ in a one-loop RPA does not allow for a closed analytic solution, for the very reason that is does *not* constitute the full scale derivative of a quantity that can be integrated explicitly. This is the very property that is cured via the Katanin replacement.

## C.4  Working at fixed density

So far we have been working at fixed chemical potential and determined the density $n = n(\mu)$ as a function thereof. But often we wish to treat the case of fixed density, the chemical potential to be determined, or at least being allowed to vary. So how can we work at fixed density? Again, the toy model offers explicit ways to do look into this. Imagine we crank up the interaction starting from the non-interacting case, as reflected in the interaction flow. The self-consistency equation (C.5) provides us with a whole family of solutions for each value of $g^2 U$, where we use the continuous flow parameter $g$ of the interaction flow as we did above:

$$\Sigma_{scH}(g^2 U, \mu) = g^2 U \frac{n}{2}(\mu, \Sigma_{scH}(g^2 U)) = g^2 U D \left( \mu - \Sigma_{scH}(g^2 U) \right).$$

If we want to describe a family of systems with the same density we thus have to adjust the chemical potential as a function of $g$ to ensure that

$$(\mu_g - \Sigma_{scH}(g^2 U)) = \frac{n}{2D} = const.$$

In particular, this applies to the non-interacting case, and we trivially have

$$(\mu_g - \Sigma_{scH}(g^2 U)) = \mu_{g=0} =: \mu_0 \,.$$

We arrive at the obvious, namely that the change in self-energy has to be compensated by a change in the chemical potential. This can be viewed as a *scale-dependent* Hartree shift which we *have* to include in the bare action of the model for the purpose of the fRG flow. Note also, that we will have $\Sigma_{scH}(g) = g\Sigma_g$ to match the respective definitions. We thus generalise $\mu \to \mu_g$ and follow the fRG recipe:

$$G_{0,g} = \frac{g}{i\omega_n - \epsilon_{\mathbf{k}} + \mu_g} \,,$$

$$G_g^{-1} = G_{0,g}^{-1} - \Sigma_g = \frac{i\omega_n - \epsilon_{\mathbf{k}} + \mu_g}{g} - \Sigma_g \,,$$

$$i.e. \quad G_g = \frac{g}{i\omega_n - \epsilon_{\mathbf{k}} + \mu_g - g\Sigma_g} \,.$$

**Note:** The introduction of a scale-dependent chemical potential alters and extends the original "regulator"-dependence of the bare action, as defined in the original interaction flow method [11]. Thus, we need to validate that the scaling properties remain valid, as we did for the QP-fRG equations in section B.3. This can be done by following the procedure in Appendix A of [21] or similar to section B.3. As discussed above, it is not guaranteed that the scaling property of the interaction flow remains valid for *arbitrary* choices of the function $\mu_g = \mu(g)$. One condition is set by the fact that $\mu$ appears in direct conjunction with $g\Sigma_g$ through all propagators, thus it needs to fulfil the very same scaling properties as $g\Sigma_g$.

We continue with the same steps as in the previous section and get

$$\dot{G}_g = \frac{d}{dg} \frac{g}{i\omega_n - \epsilon_{\mathbf{k}} + \mu_g - g\Sigma_g}$$

$$= \frac{1}{i\omega_n - \epsilon_{\mathbf{k}} + \mu_g - g\Sigma_g} - \frac{g}{(i\omega_n - \epsilon_{\mathbf{k}} + \mu - g\Sigma_g)^2} \frac{d}{dg}(\mu_g - g\Sigma_g).$$

A technically simple but conceptually non-trivial step now consists in implementing the condition $\mu_g - \Sigma_{scH}(g^2 U) = \mu_0 = const$ *implicitly* by *demanding* $\frac{d}{dg}(\mu_g - g\Sigma_g) = 0$. This implicit definition of $\mu_g$ yields

$$\dot{G}_g = \frac{d}{dg} \frac{g}{i\omega_n - \epsilon_{\mathbf{k}} + \mu_g - g\Sigma_g}$$

$$= \frac{1}{i\omega_n - \epsilon_{\mathbf{k}} + \mu_g - g\Sigma_g}$$

$$= \frac{1}{i\omega_n - \epsilon_{\mathbf{k}} + \mu_0}.$$

Thus, we have $\dot{G}_g = G_{0,g=1}$. That is, the differentiated full propagator at scale $g$ is identical to the free propagator without any scale dependence (!). The one-loop equation using RPA+Katanin diagrams thus yields an fRG equation for the self-energy that *looks* as if we had ignored all self-energy feedback and *as if* we were doing the non-self-consistent calculation we did before at fixed chemical potential. This is however *not* the case, and this matters when we generalise the argument. To be precise, we now solve *two* equations simultaneously, one for the self-energy and one for the chemical potential:

$$\dot{\Sigma}_g = UD(\mu_g - g\Sigma_g) = UD\mu_0,$$

$$\dot{\mu}_g = \frac{d}{dg}(g\Sigma_g).$$

The conceptual difference with respect to the non-self-consistent case is the fact that the physical chemical potential flows as well and compensates the flow of the self-energy. The non-trivial aspect of this becomes more obvious if we continue as above and again also look at the case of the standard self-energy feedback of the flow equation, when truncated after first order in the effective interaction, i.e. without the RPA+Katanin replacement. In that case, the fRG does not reproduce the self-consistent solution. We repeat the corresponding calculation,

including $\mu_g$. The scale-dependent free propagator reads

$$G_{0,g}^{-1} = \frac{i\omega_n - \epsilon_{\mathbf{k}} + \mu_g}{g},$$

$$\dot{G}_{0,g}^{-1} = -\frac{1}{g^2}(i\omega_n - \epsilon_{\mathbf{k}} + \mu_g) + \frac{1}{g}\frac{d\mu_g}{dg},$$

and the single-scale propagator $S_g$ is given as

$$S_g = -G_g \dot{G}_{0,g}^{-1} G_g$$

$$= \frac{g^2}{(i\omega_n - \epsilon_{\mathbf{k}} + \mu_g - g\Sigma_g)^2}\left(\frac{i\omega_n - \epsilon_{\mathbf{k}} + \mu_g}{g^2} - \frac{1}{g}\frac{d}{dg}\mu_g\right)$$

$$= \frac{1}{(i\omega_n - \epsilon_{\mathbf{k}} + \mu_g - g\Sigma_g)^2}\left(i\omega_n - \epsilon_{\mathbf{k}} + \mu_g - g\Sigma_g + g\Sigma_g - g\frac{d\mu_g}{dg}\right)$$

$$= \frac{1}{i\omega_n - \epsilon_{\mathbf{k}} + \mu_g - g\Sigma_g} + \frac{g(\Sigma_g - \frac{d\mu_g}{dg})}{(i\omega_n - \epsilon_{\mathbf{k}} + \mu_g - g\Sigma_g)^2}.$$

We again impose $\mu_g - g\Sigma_g = \mu_0$, which fixates the Fermi surface and with it the filling of the system we use to compute the rhs of the flow equation. Technically, the computation is then equivalent to ignoring all self-energy effects in the denominators of internal propagators. We can again read off the result by analogy with the initial calculation above and get

$$\dot{\Sigma}_g = UD(\mu_g - g\Sigma_g) - gUD\left(\Sigma_g - \frac{d\mu_g}{dg}\right)$$

$$= UD\mu_0 - gUD(\Sigma_g - \Sigma_g - g\dot{\Sigma}_g)$$

$$= UD\mu_0 + g^2UD\dot{\Sigma}_g$$

$$\implies (1 - g^2UD)\dot{\Sigma}_g = UD\mu_0$$

$$\implies \dot{\Sigma}_g = \frac{UD\mu_0}{1 - g^2UD}. \tag{C.10}$$

Noticeably, now the standard, truncated fRG equation renders an expression on the rhs of the flow equation that looks like an RPA-corrected non-self-consistent term, but with the "wrong" sign. In contrast, the *conceptually* more elaborate case of the self-consistent RPA+Katanin approach led to a much simpler and - for this toy model - even trivial equation.

What remains to look at to complete the comparison with the previous section is the case where we omit all self-energy effects on the rhs of the flow equation. We follow the route once again, with the single-scale propagator $S_g$ reading

$$S_g = \dot{G}_{0,g} = \frac{1}{i\omega_n - \epsilon_{\mathbf{k}} + \mu_g} - \frac{g\frac{d\mu_g}{dg}}{(i\omega_n - \epsilon_{\mathbf{k}} + \mu)^2},$$

and the flow of the self-energy is given as

$$\dot{\Sigma}_g = UD\mu_g + gUD\frac{d\mu_g}{dg} .$$

The condition of fixed density again reads $\mu_g - g\Sigma_g = \mu_0$ which we can insert:

$$\dot{\Sigma}_g = UD(\mu_0 + g\Sigma_g) + gUD\left(\Sigma_g + g\frac{d}{dg}\Sigma_g\right)$$

$$\implies (1 - g^2 UD)\dot{\Sigma}_g = UD\mu_0 + 2gUD\Sigma_g$$

$$\implies \dot{\Sigma}_g = \frac{UD\mu_0 + 2gUD\Sigma_g}{1 - g^2 UD} . \tag{C.11}$$

This is again similar to a previous equation, namely to equation (C.8), but with different signs on either side. This is not too surprising, since the structure of this simple model does not leave a lot of room for variations, and we actually invert the relation $n(\mu)$ to $\mu(n)$, such that we can expect some similarities in the structure of the equations. Yet, it is somewhat counterintuitive that for the case of the RPA+Katanin subset and the case of omitting self-energy feedback the roles are inverted: While RPA+Katanin yields the more involved equation for fixed $\mu$, and no self-energy feedback the simple non-self-consistent Hartree solution, it is the opposite when working at fixed density.

The standard fRG flow equation under the constraint of fixed density, as well as the version without any self-energy feedback at all, thus lead to an issue: For $g^2 UD \to 1$ the rhs in Eqs. (C.10) and (C.11) diverge. It is again instructive to look at sample plots, shown in Fig. 26. The flow of the self-energy diverges in both cases at $UD = g^2 = 1$, which means the flow of the chemical potential also diverges in the same manner, since we keep the difference between the two at a fixed value of $\mu_g - g\Sigma_g = \mu_0$. Unlike for the case of fixed chemical potential, for which the standard flow yielded a wrong but finite asymptotic value, the issue now appears in an inverted manner:

1. We cannot even *conduct* the flow beyond $UD = 1$, i.e. the bounding value is on the $U$-axis, rather than on the $\Sigma$-axis.

2. The behaviour is again unphysical, since a finite interaction at a finite density of states should not render it impossible to keep the particle number fixed.

As mentioned above, a more thorough comparison of the standard fRG equations, based on the single-scale propagator, to the Katanin replacement requires additional numerical checks beyond the truncation after first order in $U$. Thus, the message from the above findings is to keep an eye on such conceptual matters, but it does not invalidate prior approaches. It does however point to potential conceptual issues when self-energy effects are omitted and/or the Hartree term is treated as a simple energy shift, which are worth addressing. On the up side, the fact that the toy model allows to interpret the feedback-less computation as the proper way to work self-consistently at fixed density, may serve as an anchor around which we might argue in favour of the numerical paths that have been and are used in practice.

## C.5 From toy model to QP-fRG

A key aspect in relating the above results for the very simple toy model to the QP-fRG scheme consists in the fixation of the Fermi surface. The toy model shows a) how the RPA+Katanin

approximation reproduces the self-consistent Hartree solution, and b) that this turns out to be technically equivalent to using bare propagators on the rhs of the flow equation, if we want to work at fixed density. It thereby provides a useful anchor case, since it puts the more general calculations into perspective and hopefully in a useful vicinity of such an exact case. In turn, omitting self-energy feedback *ad hoc* and using the standard fRG truncation without the Katanin replacement, does not provide such an anchor case.

We should of course not be deceived by this tentative conceptual insight given by the toy model, due to its oversimplification. Many of these aspects are lost in the case of the 2dHM at finite temperature, such as the constant density of states, the sharp features at $T = 0$, etc. All we can argue is, that the QP-fRG flow conceptually extends this anchor case when keeping the Fermi surface fixed, but for the more general case of all coupled one-loop contributions to the flow of the effective interaction. As in the toy model, this is achieved by an implicitly flowing *momentum-dependent* chemical potential, with frequency-dependent effects added via the inclusion of the flow of the quasi-particle weight.

This choice of implicitly fixating the Fermi surface is an additional approximation to an already approximate machinery, needed to maintain numerical feasibility. The chemical potential, even when we allow it to flow, is of course not momentum-dependent, and in contrast to the toy model we would have to account for a deformation of the Fermi surface. That is a notoriously difficult task, though. Alternatively, we can check if the momentum-dependence of the self-energy at the Fermi level that determines the deformation is sizeable or weak. We did this for various cases and always found it to be very moderate at most, but that is not a general result and it does not include some residual frequency-independent tad-pole contributions, induced by the momentum-dependence of the effective interaction. Therefore, we also rely on information from prior works, which indicate that Fermi surface deformations can expected to be small for the bare interactions we look at [7, 10, 16, 70–73]. In this context, also a Pomeranchuk instability may occur and eventually lead to an effectively anisotropic hopping [73–77] and open Fermi surfaces. We did not present data for the corresponding susceptibility here, since this effect is not part of the focus here.

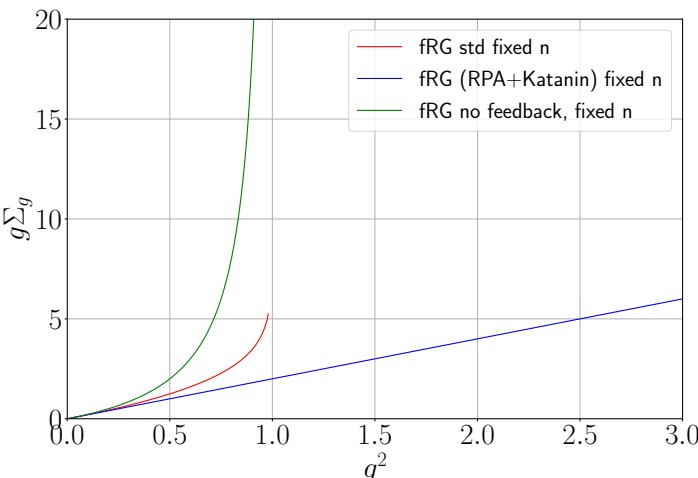

Figure 26: Comparison at fixed density of the of the self-energy flow $g\Sigma(g)$ for $UD = 1$ and $\mu = 2$ as a function of $g^2$ for the standard fRG one-loop flow, the RPA+Katanin case, and without any feedback at all. At fixed density, the RPA+Katanin flow is numerically identical to the flow without feedback at fixed chemical potential, but the conceptual interpretation differs.

While the above arguments may serve to better understand *what* we actually approximate and *how*, they are conjectures at best, and transferring the Katanin replacement to the QP-fRG scheme is not guaranteed to ameliorate the results. It is not obvious, in how far its favourable properties survive, when the effective interaction is computed by the full one-loop expression of coupled channels, required to capture the effect of competing correlation. It has been proven to cure certain fRG deficiencies and to be correct for static properties in mean-field-like situations [32], and worked better in practice for dynamical properties in zero-dimensional systems [33, 34]. But we here extend its use to dynamical properties of the normal state in a two-dimensional system, a very different situation. Also, we use a *frequency-independent* vertex and a *frequency-dependent* self-energy, a combination that it was not tailored for. For that reason, we have always included results also for the standard case without the Katanin replacement. Yet, only the Katanin replacement, together with the RPA restriction, provides a physically sound way to fix the Fermi surface in the toy model. This is not the case for the standard truncation that raises issues. Which of the two approaches is actually better in QP-fRG for the values of $U$ that are of interest needs to be cross-checked by complementary fRG implementations as well as other methods.

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
