# Peer review of "Quasi-particle functional Renormalisation Group calculations in the two-dimensional t-t'-Hubbard model"

_SciPost Physics, doi:SciPost Phys. 15, 192 (2023)_

## Round 1 · Referee Report · Anonymous (Referee 1) · 2023-5-24

Strengths

1 - The main observation of significant qualitative differences upon using the Katanin feedback is insightful and valuable to the community
2 - The work provides a thorough discussion of various different phases present in the 2D Hubbard Model for each self-energy feedback scheme.
3 - The impact of the patching granularity is explicitly investigated for each approach.

Weaknesses

1 - The 2d Hubbard model which "has in many respects not been solved to a satisfactory degree and remains a challenge" appears to be disadvantageous for comparing the effect of different approximations. Would it not be more apt to consider a simpler model, (i.e. SIAM or similar) which pose lesser challenges to numerically exact or more accurate approaches? This would also help to address one of the main questions that this work opened up: Which form of self-energy feedback can be expected to yield the qualiatively correct result.

Report

The work is original and appears correct. The author provides a very thorough analysis of the effects of different forms of self-energy feedback in the QP-fRG for different phases found in the 2D Hubbard Model. This is particularly interesting as the Katanin feedback, which is often neglected, often leads to significantly different qualitative observations. Whether the inclusion of such a feedback leads to more accurate results, however, could not be concluded.

Overall, I believe the manuscript to be suitable for publication in SciPost, once the remarks below are sufficiently addressed, as it is of interest to the community and leads up to interesting future work.

Requested changes

  • pg. 3: It is mentioned in the text that the QP-fRG provides the simplest possible way to include self energy feedback into the fRG. Please be more explicit about the challenges in including the self energy feedback in regular FRG approaches.
  • pg. 4: It would be helpful to add more motivation behind chosing the 2dHM as a benchmark system, especially given the challenges in interpreting the results (see above).
  • pg. 5: It is mentioned that the frequency dependence of the vertex is neglected. Can you comment on the impact this approximation has on the validity of the (frequency dependent) self-energy feedback? How justifiable is this approximation in general, given that several recent works focus on improvements in numerically feasible frequency representations of the vertex.
  • pg 9.: It is stated regarding the use of eigenvalues of the effective interaction that "This analysis is limited in some respects". Is it possible to be slightly more precise about what the limitations of this approach are?
  • The manuscript relies in several points on the Z-factor which is however not formally introduced. It would be helpful to add a definition and possibly a brief discussion of its relevance.

  • pg. 13, below 3.6: please correct the typo $V_max$ to $V_{max}$

Question (no change requested): - Figures: The plots of $V_{max}$ and the leading eigenvalues show discrete points along the x-axis, which is the flowing parameter and should be accessible continuously in modern RK solvers. The flow of the Z-factor, meanwhile is plotted in terms of continuous steps as expected. Why is the value of the Z-factor interpolated more often during the flow when compared to the maximum value of the vertex and its leading eigenvalues? Could these not also be computed and saved during the flow?

  • validity: high
  • significance: high
  • originality: high
  • clarity: high
  • formatting: perfect
  • grammar: perfect

Author:  Daniel Rohe  on 2023-07-31  [id 3855]

(in reply to Report 1 on 2023-05-24)

Warnings issued while processing user-supplied markup:

  • Inconsistency: Markdown and reStructuredText syntaxes are mixed. Markdown will be used.
    Add "#coerce:reST" or "#coerce:plain" as the first line of your text to force reStructuredText or no markup.
    You may also contact the helpdesk if the formatting is incorrect and you are unable to edit your text.

Reply:

I thank both referees for investing their time and effort to provide very constructive and valuable feedback! I acknowledge all comments and hope that I understood them correctly.

I have tried to properly address all issues and questions. Please find the specific answers below, inline with the original comments/questions. I hope the changes made in the manuscript comply with the intentions of the referees.

---------- Original Report 1 plus answers inline -----------

Strengths

1 - The main observation of significant qualitative differences upon using the Katanin feedback is insightful and valuable to the community 2 - The work provides a thorough discussion of various different phases present in the 2D Hubbard Model for each self-energy feedback scheme. 3 - The impact of the patching granularity is explicitly investigated for each approach.

Weaknesses

1 - The 2d Hubbard model which "has in many respects not been solved to a satisfactory degree and remains a challenge" appears to be disadvantageous for comparing the effect of different approximations. Would it not be more apt to consider a simpler model, (i.e. SIAM or similar) which pose lesser challenges to numerically exact or more accurate approaches? This would also help to address one of the main questions that this work opened up: Which form of self-energy feedback can be expected to yield the qualitatively correct result.

Reply/comment:

We agree with the referee that the question of the "best" type of self-energy feedback constitutes a valid focus on its own. In lower dimensions, this has been done to a certain extent in previous works, as cited in the first version of the manuscript just before section 3 (Hedden 10.1088/0953-8984/16/29/019 , Karrasch 10.1088/0953-8984/20/34/345205). This has served as an indicator for the 2d case, and we have extended the wording of the relevant citation to make this clearer. At the same time, the QP-fRG method relies on a Fermi-liquid-like description, and as outlined in the preceding paper (doi:10.21468/SciPostPhys.9.6.084) this description resides on the very edge of applicability in two dimensions. In lower dimensions an FL-based approach is at least questionable, if not invalid per se. Thus, even if more robust conclusions on fRG schemes could be drawn from cases in lower dimensions, they do not need to carry over to two dimensions.

For the work presented here, the main motivation is the analysis of a potentially improved access to the properties of the 2dHM, being aware of the fact that it is hard to know what is "better". The observation that the Katanin replacement can lead to qualitative changes was actually not expected and emerged as a new result. For the half-filled nested case it appears to improve the cross-over scales, and the questions was if this carries over to the more general case.

Some of these aspects served as a motivation for the discussion in appendix section B.5 of the manuscript.

Report

The work is original and appears correct. The author provides a very thorough analysis of the effects of different forms of self-energy feedback in the QP-fRG for different phases found in the 2D Hubbard Model. This is particularly interesting as the Katanin feedback, which is often neglected, often leads to significantly different qualitative observations. Whether the inclusion of such a feedback leads to more accurate results, however, could not be concluded.

Overall, I believe the manuscript to be suitable for publication in SciPost, once the remarks below are sufficiently addressed, as it is of interest to the community and leads up to interesting future work.

Requested changes

i) - pg. 3: It is mentioned in the text that the QP-fRG provides the simplest possible way to include self energy feedback into the fRG. Please be more explicit about the challenges in including the self energy feedback in regular FRG approaches.

Response:

This seems to be a misunderstanding due to an ambiguous formulation in the manuscript: What is meant by "albeit simplest means" is the conceptual way of implementing self-energy feedback, i.e. via a quasi-particle weight. The numerical effort is still considerable and not the (only) reason for its choice in comparison to other schemes. The paragraph was rephrased. As for other schemes in which self-energy feedback is included, they are addressed in the discussion after the results section. Concerning their numerical effort and the challenges, we cannot properly judge on them since that depends on many details and we do not know the specific ways in which they have been implemented.

As a remark concerning numerical challenges: For some fRG codes parallel versions do exist, for some not or it is not clear. To efficiently harvest that potential is a complex subject on its very own, as has been addressed in prior works (DOI 10.1016/j.cpc.2016.12.013 , 10.1016/j.cpc.2016.05.024). Finding the "best" choice of an fRG scheme for a specific application of interest also includes this aspect. The QP-fRG method was implemented in a way that scales up to a nearly arbitrary number of nodes by combining MPI and OpenMP. Considering the ongoing change and evolution of HPC architectures, parallelising fRG codes to scale efficiently is an ongoing task.

ii) - pg. 4: It would be helpful to add more motivation behind chosing the 2dHM as a benchmark system, especially given the challenges in interpreting the results (see above).

Answer: A paragraph on this aspect was added as an intro in the section "Model and Method" , c.f. also the comment above on "weaknesses". I could imagine that the answer and the changes are not quite satisfactory to the referee. I very much agree that the 2dHM is hardly a well-defined benchmark case in the proper sense, both numerically and regarding the physics. Yet, the path towards better methods for the search for the "true" physics of the model includes steps like this, which may (hopefully) serve in some way for this very purpose.

iii) - pg. 5: It is mentioned that the frequency dependence of the vertex is neglected. Can you comment on the impact this approximation has on the validity of the (frequency dependent) self-energy feedback? How justifiable is this approximation in general, given that several recent works focus on improvements in numerically feasible frequency representations of the vertex.

Answer: This is now addressed in a new section 3 on "Remarks and limitations" - c.f. also comment 1 in report 2 and the extended answer to it.

iv) - pg 9.: It is stated regarding the use of eigenvalues of the effective interaction that "This analysis is limited in some respects". Is it possible to be slightly more precise about what the limitations of this approach are?

Answer: The sentence was extended. Also, a reminder was added at the end of the section that this type of fRG method cannot detect true phase transitions but rather the onset of strong correlations. Critical scales, in case they emerge, have to be seen as mean-field-like.

v) - The manuscript relies in several points on the Z-factor which is however not formally introduced. It would be helpful to add a definition and possibly a brief discussion of its relevance.

Answer: We rearranged the part on self-energy results, see also the reply to point iii). Within that part we also included an explicit mentioning of the definition of Z, see the new section 3.1

(C.f. also the suggestion to rearrange the manuscript in report 2 and the reply to it.)

vi) - pg. 13, below 3.6: please correct the typo Vmax to Vmax

Answer: Done

Question (no change requested):

  • Figures: The plots of Vmax and the leading eigenvalues show discrete points along the x-axis, which is the flowing parameter and should be accessible continuously in modern RK solvers. The flow of the Z-factor, meanwhile is plotted in terms of continuous steps as expected. Why is the value of the Z-factor interpolated more often during the flow when compared to the maximum value of the vertex and its leading eigenvalues? Could these not also be computed and saved during the flow?

Reply: There is no difference in the resolution of either data. Symbols are not included in the plots for Z to facilitate the visual perception, since otherwise there would be cluttering. The other plots include symbols to indicate the points for which raw data was computed, and this information also applies to the graphs for Z. The interpolation is then done by the plotting software, not by the RK solver.

---

## Round 1 · Referee Report · Andrey Katanin (Referee 2) · 2023-6-8

Strengths

The paper "Quasi-particle functional renormalization group..." addresses the problem of an importance of the self-energy corrections in fRG flow and their effect in various schemes. The proper treatment of the self-energy corrections is an important aspect of the renormalization group procedure beyond the 1-loop approach. It is "responsible" for the renormalization of quasiparticles, their damping, as well as splitting/changing of electronic spectra in the symmetry broken phases. Therefore, this topic is important and worth consideration. The paper considers the effect of the self-energy corrections within the earlier proposed quasi-particle functional renormalization group scheme, treating self-energies via the quasi particle residues. This essentially simplifies calculations and makes them feasible for broad set of parameters.

Weaknesses

At the same time, I have questions regarding the range of applicability of considered approach (see below), which in my opinion is worth reflecting in the paper.

Also, I believe that restructuring of the paper may be useful for better readability, see suggestions below.

Report

As I mention above, the quasi-particle functional renormalization group approach greatly simplifies the calculations. However, I worry about the following points:

  1. Keeping the Z-factors only violates the normalization of the total electron spectral weight at each k-point to one, the weight becomes Z<1 instead. Where (in which situations) this approach is expected to be applicable? Does not it loose some important incoherent contributions to the spectral functions? In particular, in the vicinity of the symmetry-broken phases the self-energy in known to develop an anomalous atomic-like frequency dependence, \Sigma(nu)\propto U^2/nu, see, e.g.Phys. Rev. B 71, 085105 (2005), Phys. Rev. B 72, 035111 (2005) and references therein. This dependence is not represented by a single quasiparticle weight. How does that affects the flow? Can that be the origin of the absence of ferromagnetic instability in the considered approach at large t'?

  2. Possibly related to the question 1 above. I am somewhat surprised that Katanin scheme leads to the strongest suppression of vertices and instabilities. Indeed, this scheme modifies fRG equations in such a way that they becomes closer to the mean-field ones. I.e. it suppresses fluctuations. This is confirmed by the fact that in the reduced BCS approach (M. Salmhofer, et. al., Prog. Theor. .Phys. 112, No. 6 (2004) 943) the superconducting gap increases from 0.75 BCS gap to the 1 BCS gap when this scheme is applied. Therefore, naively, I would expect that this scheme somewhat enhances divergence of the vertices in the symmetric phase. It would be helpful to address the question why does not this logic work. Is it related to the quasi-particle approximation or something else?

  3. The set of parameters considered in the paper does not contain van Hove filling, t'=-0.2 (or close values). This describes the physics of cuprates, and it would be interesting to see what happens in these situations. Regrading the case of van Hove filling at t'=-0.28, where the Katanin scheme leads to qualitative change of an instability, one should remember that this t' is already on the verge of AFM-FM phase transition, and it might be that the self-energies effects shift the system towards this transition. Could that be the origin of qualitative change of the behaviour?

  4. Related to the questions 1,3 above. What happens with the filling in the quasi-particle functional renormalization group at fixed mu? Can it be correctly estimated in view of the loss of the spectral weight mentioned in p. 1 above? In Appendix B the author addresses the flow of the fillings, but only in the scheme with the full self energy in the denominators of Green's function. This consideration is very interesting, but how is that related to the main text of the paper?

Requested changes

I suggest the author to address in the paper pp. 1-4 above.

Apart from that, I think it would be helpful to integrate the content of Sects. 5, A.2, A.3 into the main text of the paper. In particular, it is worth to include figures together with their discussions.

Minor points:
Stoner/Thouless is probably better named as Slater/Thouless, since the author considers AFM instability, not the FM one.
Quasi half filling is probably better refer as proximity to half filling.
The denominator in Eqs. (7) and in the following is related to the Kanamori screening. Indeed, differentiating Eq. in the end of Sect. B.2 over mu one finds the compressibility 2D/(1+U*D) which is just the Kanamori result for constant density of states. Therefore, the reference to Kanamori paper would be helpful.
The value of temperature can be removed from the names of Sects. 5.3-5.6 since it is kept constant.

  • validity: good
  • significance: high
  • originality: high
  • clarity: good
  • formatting: good
  • grammar: perfect

Author:  Daniel Rohe  on 2023-07-31  [id 3856]

(in reply to Report 2 by Andrey Katanin on 2023-06-08)

Warnings issued while processing user-supplied markup:

  • Inconsistency: plain/Markdown and reStructuredText syntaxes are mixed. Markdown will be used.
    Add "#coerce:reST" or "#coerce:plain" as the first line of your text to force reStructuredText or no markup.
    You may also contact the helpdesk if the formatting is incorrect and you are unable to edit your text.

Reply:

I thank both referees for investing their time and effort to provide very constructive and valuable feedback! I acknowledge all comments and hope that I understood them correctly.

I have tried to properly address all issues and questions. Please find the specific answers below, inline with the original comments/questions. I hope the changes made in the manuscript comply with the intentions of the referees.

---------- Original Report 2 plus answers inline -----------

Strengths

The paper "Quasi-particle functional renormalization group..." addresses the problem of an importance of the self-energy corrections in fRG flow and their effect in various schemes. The proper treatment of the self-energy corrections is an important aspect of the renormalization group procedure beyond the 1-loop approach. It is "responsible" for the renormalization of quasiparticles, their damping, as well as splitting/changing of electronic spectra in the symmetry broken phases. Therefore, this topic is important and worth consideration. The paper considers the effect of the self-energy corrections within the earlier proposed quasi-particle functional renormalization group scheme, treating self-energies via the quasi particle residues. This essentially simplifies calculations and makes them feasible for broad set of parameters. Weaknesses

At the same time, I have questions regarding the range of applicability of considered approach (see below), which in my opinion is worth reflecting in the paper.

Also, I believe that restructuring of the paper may be useful for better readability, see suggestions below. Report

As I mention above, the quasi-particle functional renormalization group approach greatly simplifies the calculations. However, I worry about the following points:

  1. Keeping the Z-factors only violates the normalization of the total electron spectral weight at each k-point to one, the weight becomes Z<1 instead. Where (in which situations) this approach is expected to be applicable? Does not it loose some important incoherent contributions to the spectral functions? In particular, in the vicinity of the symmetry-broken phases the self-energy in known to develop an anomalous atomic-like frequency dependence, \Sigma(nu)\propto U^2/nu, see, e.g.Phys. Rev. B 71, 085105 (2005), Phys. Rev. B 72, 035111 (2005) and references therein. This dependence is not represented by a single quasiparticle weight. How does that affects the flow? Can that be the origin of the absence of ferromagnetic instability in the considered approach at large t'?

Brief answer:

We added section 3 on "Remarks and limitations", also to follow the suggestion by the referee to reorganise the structure. As part of this we tried to properly address these points in this new section, also covering remark v) in report 1.

Extended answer:

The spectral weight on internal lines on the rhs of the fRGE is indeed reduced, reflecting the fact that the coherent contributions of quasi-particles are affected by a reduction in their weight, while the incoherent background being smeared out is neglected, by the argument that it does not influence the leading divergent behaviour of the effective interaction. This becomes indeed invalid if other coherent-like structures with significant weight appear in the spectral function to be approximated, as also discussed in doi 10.21468/SciPostPhys.9.6.084. The resulting full spectral function, however, as calculated from the resulting self-energy, remains normalised to unity, at all fRG scales. In other words, the quasi-particle weight acts as a filter on the rhs of the fRGE that extracts the coherent parts of the contributions. A more direct and explicit numerical comparison to other schemes which work with the full functional form of the self-energy and the full spectral function also on internal lines, even if it is on the imaginary axis, would indeed be desirable for future works to study the deviations due to neglecting the (mostly) incoherent parts.

Regarding atomic-like structures, as referred to in the works cited above, they are to my understanding mainly found to be relevant BELOW some cross-over scale T^, and possibly also in a small region above, at weak to moderate bare interaction. As also stated above, in case such a structure appears in the spectral function or even fully replaces the FL structure, the QP-fRG flow based on the Fermi-liquid description becomes invalid, as also discussed in doi 10.21468/SciPostPhys.9.6.084. Here, as in many fRG treatments, the validity is restricted to T>T^, where the divergences in the effective interaction are mean-field like. For some cases in the QP-fRG, the inclusion of self-energy effects using the Katanin replacement can avoid this divergence. In that case, the fRG method is limited by the fact that it is also perturbative in the bare interaction. Thus, while it is tempting to extend the interpretation of the results to the physical case of larger bare couplings, this is a priori of limited validity, and we do not want to speculate too much beyond that point. We tried to make this clear in the manuscript. That said, we do not see that the absence of a Ferromagnetic instability is related to this. However, we would anticipate that it might be restored when moving further into the respective region of the phase diagram, similar to the fact that the AFM instability prevails also under the Katanin replacement close enough to perfect nesting at van Hove filling. This would be an analysis that can be conducted in future work.

An important aspect which is related to the comment is the role (or not) of the frequency dependence of the effective interaction, in particular in cases where the divergence of the effective interaction disappears. We do restrict the interpretation of the corresponding results to values of $U<4$. In that region, given that the Slater/Thouless enhancement remains small, we would expect the frequency-dependence to also be of non-singular type and comparatively small. Beyond that bare interaction strength, it may well be that such a dependence builds up and leads to additional coherent structures such as those mentioned by the referee. It would therefore be interesting to complement the numerical setup by a similar implementation on the imaginary axis and including a frequency dependence of both vertex and self-energy. It would be important to keep all other parameters and choices identical, where possible. A small step in that direction has been taken (Reference [41] in the first version of the manuscript]), but it is uncertain if this will happen beyond first attempts.

To add another aspect related to this: If one hypothetically cranks up the bare interaction in second order perturbation theory even further, the well-known isolated "atomic-like" poles appear in the spectral function below and above the band edges. One could expect this to happen already for smaller values of $U$ in the QP-fRG flow, due to a growth or divergence of the effective interaction. However, we never observed a strong renormalisation of the scale where this happens, at least not for the given stopping criterion.

  1. Possibly related to the question 1 above. I am somewhat surprised that Katanin scheme leads to the strongest suppression of vertices and instabilities. Indeed, this scheme modifies fRG equations in such a way that they becomes closer to the mean-field ones. I.e. it suppresses fluctuations. This is confirmed by the fact that in the reduced BCS approach (M. Salmhofer, et. al., Prog. Theor. .Phys. 112, No. 6 (2004) 943) the superconducting gap increases from 0.75 BCS gap to the 1 BCS gap when this scheme is applied. Therefore, naively, I would expect that this scheme somewhat enhances divergence of the vertices in the symmetric phase. It would be helpful to address the question why does not this logic work. Is it related to the quasi-particle approximation or something else?

Answer: A discussion on this is now offered in the revised manuscript in section 5 under "Caveats, limitations and additional remarks". I however cannot give a conclusive answer to this. I assume the apparent discrepancy stems from a conjecture to BCS theory, where T_c is proportional to the gap in the ground state, and thus for a smaller gap in the ground state one could expect also a smaller transition temperature. Yet, that relation involves some underlying prefactors and conditions which may differ for the two cases.

Remark: At the technical level of equations it is possible to identify the reason why the Katanin replacement can lead to saturation effects at all, as described in (doi 10.21468/SciPostPhys.9.6.084).

  1. The set of parameters considered in the paper does not contain van Hove filling, t'=-0.2 (or close values). This describes the physics of cuprates, and it would be interesting to see what happens in these situations. Regrading the case of van Hove filling at t'=-0.28, where the Katanin scheme leads to qualitative change of an instability, one should remember that this t' is already on the verge of AFM-FM phase transition, and it might be that the self-energies effects shift the system towards this transition. Could that be the origin of qualitative change of the behaviour?

In a way my answer would be "yes" and "no": We have in addition to the first version conducted the computation and added Figures and a discussion for the case t'=-0.2 at van Hove filling, as a separate set of results. In fact, most aspects remain similar to the case t'=-0.28 and we do not see any obvious indication that the self-energy "shifts" the system towards the FM-transition in that very sense. However, the saturation effect is indeed much more evident for t'=-0.28, since this is a priori closer to the FM-transition which suppresses the scales already for the case without feedback. The issue that prevails is the question whether for t'=-0.2 the saturating behaviour converges as a function of the patching granularity or not. In essence, it appears that for true saturation (in the technical sense) one either has to move far enough away from vH filling, or increase t' beyond a certain value - or both, as discussed in the section on the role of granularity. For the case t'=-0.2 at van Hove filling this is not obvious.

  1. Related to the questions 1,3 above. What happens with the filling in the quasi-particle functional renormalization group at fixed mu? Can it be correctly estimated in view of the loss of the spectral weight mentioned in p. 1 above? In Appendix B the author addresses the flow of the fillings, but only in the scheme with the full self energy in the denominators of Green's function. This consideration is very interesting, but how is that related to the main text of the paper?

A comment on this was added in the section on the method, in item (v) of the second list, on page 6 of the revised manuscript. I cannot cover it in more detail, since this approach is not feasible in the QP-fRG setup for various reasons:

a) it would require an additional step of computing the scale-dependent density. The resolution of the patching as well as the QP-approximation prevent a proper integration of such a step.

b) Alternatively, when working with the frequency-independent part of the self-energy to determine the flow of the Fermi surface, it would require the additional calculation of the original one-loop diagram, disentangled from all two-loop contributions in the second derivative. While this is very involved but might be feasible, working at constant \mu would above all

c) continuously shift the Fermi surface as a function of the fRG scale parameter, which in turn necessitates a dynamical adjustment of the patching and the parametrisation, which is (currently) infeasible in practical terms within this setup.

The toy model, in contrast, has the very advantage that it allows for both views, due to the simplicity of the model for which the analytic treatment can be carried out to a much larger extent than for the 2dHM.

We note that in other fRG schemes the Fermi surface - and with it the filling - does actually flow, see e.g. DOI: 10.1103/PhysRevResearch.2.033372. But even for this case, the Hartree contribution is implicitly subtracted, i.e. the chemical potential is not really held fixed, but the density varies also (away from half filling). There, it is also recognised that a comparison between methods and their results can be subtle due to this very effect.

Requested changes

I suggest the author to address in the paper pp. 1-4 above.

Apart from that, I think it would be helpful to integrate the content of Sects. 5, A.2, A.3 into the main text of the paper. In particular, it is worth to include figures together with their discussions.

Comment:

We moved the part on results for SOPT from A.3 to a new section 3.2 in the main body and fused them with the inclusion of a request from Report 1. As for the definition of the Eigenvalues I prefer to keep them in the appendix.

Minor points:

i) Stoner/Thouless is probably better named as Slater/Thouless, since the author considers AFM instability, not the FM one.

Done

ii) Quasi half filling is probably better refer as proximity to half filling.

Done

iii) The denominator in Eqs. (7) and in the following is related to the Kanamori screening. Indeed, differentiating Eq. in the end of Sect. B.2 over mu one finds the compressibility 2D/(1+U*D) which is just the Kanamori result for constant density of states. Therefore, the reference to Kanamori paper would be helpful.

We added a mentioning of the Kanamori paper at the end of section B.2. Minor comment to (hopefully) avoid confusion: The compressibility is usually defined as (1/n^2)*(dn/d\mu) (c.f. e.g. Negele/Orland, Pines/Nozière). Here, this gives (1+UD)/(2D\mu^2). It changes sign (as a function of U) at 1+UD = 0, signalling the instability.

Done

iv) The value of temperature can be removed from the names of Sects. 5.3-5.6 since it is kept constant.

Done

---

## Round 2 · Referee Report · Anonymous (Referee 1) · 2023-8-9

Report

I thank the authors for adequate clarification of my comments and for improved discussion of the rationale for the chosen model and the underlying approximations in the manuscript.
The drastic differences that result from the inclusion of self-energy feedback is an interesting observation. I hope this can be explored further, although I agree with the author that identifying a fully rigorous benchmarking model is a challenge in itself.
Since the changes made in the resubmission address my concerns to my satisfaction, I recommend publication of the manuscript.

---

## Round 2 · Referee Report · Andrey Katanin (Referee 2) · 2023-8-15

Strengths

In the revised paper the author has introduced the changes according to the Referees' suggestions. The revised paper can be therefore published .

Weaknesses

-

Report

The revised paper meets acceptance criteria.

Requested changes

-

---

## Round 2 · List of Changes

Addressed comments by referees, including:

  • added a new section 3 on remarks and limitations
  • added section 5.2 on caveats
  • rearranged former appendix A.3 into main body
  • various smaller updates

---

## Editorial Decision

published